# Trans-driven variation in expression is common among detoxification genes in the extreme generalist herbivore *Tetranychus urticae*

**Andre H. Kurlovs**[1,2☯], **Berdien De Beer**[1☯], **Meiyuan Ji**[2☯], **Marilou Vandenhole**[1], **Tim De Meyer**[3], **René Feyereisen**[1,4], **Richard M. Clark**[2,5]*, **Thomas Van Leeuwen**[1]*

**1** Department of Plants and Crops, Faculty of Bioscience Engineering, Ghent University, Ghent, Belgium, **2** School of Biological Sciences, University of Utah, Salt Lake City, Utah, United States of America, **3** Department of Data Analysis and Mathematical Modelling, Faculty of Bioscience Engineering, Ghent University, Ghent, Belgium, **4** Department of Plant and Environmental Sciences, University of Copenhagen, Thorvaldsensvej, Copenhagen, Denmark, **5** Henry Eyring Center for Cell and Genome Science, University of Utah, Salt Lake City, Utah, United States of America

☯ These authors contributed equally to this work.
* clark@biology.utah.edu (RMC); Thomas.VanLeeuwen@UGent.be (TVL)

**Data Availability Statement:** DNA-seq and RNA-seq read data have been deposited to NCBI under identifiers PRJNA799176 and PRJNA801103,

## Abstract

The extreme adaptation potential of the generalist herbivore *Tetranychus urticae* (the two-spotted spider mite) to pesticides as well as diverse host plants has been associated with clade-specific gene expansions in known detoxifying enzyme families, and with extensive and rapid transcriptional responses. However, how this broad transcriptional potential is regulated remains largely unknown. Using a parental/F1 design in which four inbred strains were crossed to a common inbred strain, we assessed the genetic basis and inheritance of gene expression variation in *T. urticae*. Mirroring known phenotypic variation in the progenitor strains of the inbreds, we confirmed that the inbred strains we created were genetically distinct, varied markedly in pesticide resistance, and also captured variation in host plant fitness as is commonly observed in this species. By examining differences in gene expression between parents and allele-specific expression in F1s, we found that variation in RNA abundance was more often explained in *trans* as compared to *cis*, with the former associated with dominance in inheritance. Strikingly, in a gene ontology analysis, detoxification genes of the cytochrome P450 monooxygenase (CYP) family, as well as dioxygenases (DOGs) acquired from horizontal gene transfer from fungi, were specifically enriched at the extremes of *trans*-driven up- and downregulation. In particular, multiple CYPs and DOGs with broad substrate-specificities for pesticides or plant specialized compounds were exceptionally highly upregulated as a result of *trans*-regulatory variation, or in some cases synergism of *cis* and *trans*, in the most multi-pesticide resistant strains. Collectively, our findings highlight the potential importance of *trans*-driven expression variation in genes associated with xenobiotic metabolism and host plant use for rapid adaptation in *T. urticae*, and also suggests modular control of these genes, a regulatory architecture that might ameliorate negative pleiotropic effects.

respectively. Pairwise strain variant call data, and read count data, have been depos-ited to figshare (https://doi.org/10.6084/m9.figshare.20291175. v1). The additional data are available as supplementary tables, and code used in the study is available at Github (https://github.com/akurlovs/ase).

**Funding:** This work was supported by the Research Council (ERC) under the European Union's Horizon 2020 research and innovation program (https://cordis.europa.eu/en), grant 772026-POLYADAPT and 773902–SuperPests, and Bijzonder Onderzoeksfonds UGent (BOFSTA2017003701) (https://www.ugent.be/nl/onderzoek/financiering/bof) to TVL. The funders had no role in study design, data collection and analysis, decision to publish, or preparation of the manuscript.

**Competing interests:** The authors have declared that no competing interests exist.

## Author summary

The two-spotted spider mite, *Tetranychus urticae*, is a generalist herbivore and pest of diverse crops globally. In response to the plethora of chemicals used for its control, the species rapidly evolves pesticide resistance. Further, experimental evolution studies with *T. urticae* populations have demonstrated adaptation to challenging host plants in as few as five generations. The adaptation of *T. urticae* to pesticides and host plants has been associated with large transcriptome changes, including for genes associated with detoxi-fication of pesticides and toxic plant compounds. Nevertheless, the basis of the observed transcriptome variation has remained largely unknown. Here, we examined the genetic control and inheritance of expression differences among five inbred *T. urticae* strains, including several with histories of intense pesticide selection. With a parental/F1 experi-mental design, we found that *trans* effects were common in explaining variation in detoxi-fication gene expression, with the *trans*-driven upregulation of a subset of cytochrome P450 monooxygenases of broad substrate specificity especially striking in the most pesti-cide resistant strains. Our findings suggest that genetic variation acting with dominant or additive inheritance to impact the regulation of modules of detoxification genes may be an important target of selection during rapid pesticide and host plant evolution in herbivores.

## Introduction

Genetic variation as a requisite for evolutionary change through selection, recombination and drift lies at the basis of our current understanding of evolution [1,2]. Nucleotide changes in coding sequences were among the first to be associated with phenotypic differences of adaptive significance. Subsequently, advances in genomic methods have facilitated a broader under-standing of how other types of coding sequence variation–such as structural or copy number variation–impact phenotypes [3,4]. Likewise, a number of specific instances of regulatory vari-ation in genic regions impacting gene expression phenotypes of adaptive relevance have been characterized in detail [5–7]. More recently, advances in experimental designs and methods have allowed the genetic control of gene expression variation to be assessed genome-wide [8,9], offering the possibility of a more comprehensive understanding of how such variation arises and is selected upon to impact phenotypes [10].

One widely used approach to understand the genetic control of gene expression variation in diploid organisms, as well as its associated inheritance, involves comparing expression between parents and F1 offspring [11–13]. If a difference in gene expression between two parental inbred strains is due to *cis* effects only (i.e., changes in promoter or enhancer regions that influence gene expression on the same chromosome), the same difference ratio will be observed for the respective parent-of-origin transcripts in F1 offspring (i.e., allele-specific expression, or ASE; parent-of-origin for transcripts can be assigned using RNA-seq when vari-ants are present in transcribed sequences). However, if the ratios for expression differences between parents are not the same as for F1 ASE ratios, *trans* effects–which reflect variation in diffusible factors such as transcription factors or components of upstream signaling pathways that can impact the expression of multiple target genes on both chromosomes–contribute to expression variation [8]. This type of parental/F1 design also enables the degree of domi-nance–an important factor in understanding responses to selection–to be inferred [12–15]. A number of studies in diploid organisms, often using genetic models selected in part because of

their extensive genomic resources, have revealed trends about the frequency of *cis* and *trans* effects, their interactions, as well as associations with inheritance type. For example, in many instances, expression variation more often results from variation in *trans* regulation compared to *cis* regulation in intraspecific comparisons, with the opposite pattern observed for interspecific ones [11]. Further, for genes for which *cis* effects explain expression variation, additivity in inheritance is often observed; in contrast, genes for which variation is explained in *trans* are usually enriched for dominance [12–16].

However, despite the resolution and comprehensive nature of parental/F1 and related experimental designs, connecting the genetic variation that impacts gene expression to adaptation has remained challenging. In part, this results from a lack of understanding of underlying selective forces. A classic example of rapid adaptation for which the agent(s) are known explicitly is pesticide resistance evolution. This phenotype has been studied extensively in arthropods, including in herbivores [17–20]. For the herbivorous insects and mites that consume plant tissue, chemical control by pesticides is widely used. Resistance evolution is a major obstacle for agriculture, and understanding inheritance (dominance or lack thereof) can be critical to inform approaches for resistance management [21–23]. Further, albeit with less precision in space, time, or nature of the selective agent(s), adaptation of herbivores to their host plants has also provided well-known examples of evolution in action [24,25]. Finally, a number of gene families have been associated with the detoxification, sequestration, or transport and excretion of pesticides, and are either known or are likely to be critical in host plant adaptation as well (i.e., to overcome toxic plant-produced specialized compounds, although plants deploy other defense strategies as well) [24–32]. Therefore, many selective agents are known for adaptation in herbivores, sometimes exactly (pesticides), as are genes and gene families that are strong candidates to underlie rapid phenotypic evolution.

In this study, we have determined the genetic basis and inheritance of gene expression variation among strains of the two-spotted spider mite, *Tetranychus urticae*. This generalist (polyphagous) mite is at the extreme end of the generalist-to-specialist spectrum, as it has been documented to feed on a staggering 1,100 plant species from more than 140 plant families, including more than 150 crops [33]. In addition, *T. urticae* is well-known for developing acaricide resistance in as fast as a few years after the introduction of most acaricides used for its control (acaricides are pesticides that are active against mites, which belong to the Acari within Arthropoda subphylum Chelicerata) [19]. The extreme adaptation potential of this species has been associated with clade-specific gene expansions in known detoxifying enzyme families, such as cytochrome P450 monooxygenases (CYPs), mu-class glutathione-S-transferases (GSTs), a new clade of carboxyl-choline esterases (CCEs), and an unexpectedly large repertoire of ATP-binding cassette (ABC) and major facilitator superfamily (MFS) transporters [25,27,33]. In addition, it acquired several putative detoxification genes by horizontal gene transfer that subsequently proliferated in the genome [34]. These include intradiol-ring cleavage dioxygenases (DOGs) that have recently been shown to cleave an unexpectedly wide range of aromatic plant defense specialized compounds [32], as well as bacterial UDP-glucuronosyl-transferases (UGTs) [35], demonstrating the adaptive advantage of horizontal gene transfer in the context of detoxification and host plant use [33,34,36]. Gene-expression studies have revealed large and similar transcriptomic differences in these gene families between acaricide susceptible and resistant *T. urticae* strains, as well as after adaptation to new host plants [24,25,37], suggesting a link between host plant use and acaricide resistance. The extensive and rapid transcriptional reprogramming associated with the exceptional adaptation potential of *T. urticae* raises several questions about the mechanisms of gene regulation in polyphagous herbivores. Given the often very large population sizes of this species in agricultural settings, selection acting on *cis* variants at many loci is plausible. However, the often coordinated and

rapid responses in gene expression patterns observed between strains after selection by both acaricides and host plants [24,25] raises the alternative possibility of selection acting on *trans*-acting factors that serve as master regulators of many genes in detoxification/adaptation pathways that are located throughout the genome.

As opposed to many herbivores, *T. urticae* has a short life cycle and a high-quality genome [33]. Further, construction of inbred (isogenic) strains by sequential rounds of mother-son crosses is straightforward in this haplodiploid species (males are haploid and females are diploid) [38]. Therefore, *T. urticae* is optimally suited for a study relying on F1s from parental inbred strain crosses to determine *cis/trans* contributions to and inheritance of gene expression. Here, we selected four genetically divergent *T. urticae* strains with different evolutionary histories, including two highly multi-acaricide resistant strains, and inbred them. We then crossed the inbred strains to a common inbred parental strain that was comparatively susceptible to most acaricides tested. Using parental/F1 experimental designs, we found that extensive expression differences in diploid females among strains were often controlled in *trans*. This included subsets of genes in multiple detoxification gene families, indicative of modular control of detoxification and host plant associated pathways in this generalist mite pest. Notably, *trans* control contributed prominently to the exceptionally high expression of a clade of CYPs with broad substrate specificity in the acaricide resistant strains.

## Results

### Five inbred strains harbor extensive genetic variation

Starting with a single virgin female from each of five outbred strains previously known to vary in acaricide resistance [25,39,40], we performed inbreeding with mother-son crosses for at least seven generations (see Methods). Genome sequencing and variant prediction for the subsequent five inbred strains, denoted MR-VPi, MAR-ABi, JP-RRi, ROS-ITi, and SOL-BEi, revealed 1.19 million high-quality SNPs and small indels among strains (a variant every ~75 bp in the 90 Mb genome [33]), a level of polymorphism consistent with prior studies with *T. urticae* [41–44]. Further, more than 98% of SNP sites were fixed in each of the five strains, confirming earlier reports that ~7 generations of inbreeding in this haplodiploid species results in isogenic (or nearly isogenic) strains [41]. To assess genetic relationships among strains, we performed a principal component analysis (PCA) using the SNP data (Fig 1A). While the inbreeding of the five strains was performed as part of the current study, the SOL-BEi and MAR-ABi strains, and respective genomic sequence and parental RNA-seq data generated for these strains, was used by De Beer et al. [45] in a bulk segregant analysis (BSA) study that identified two quantitative trait loci (QTL) for resistance to the acaricide fenbutatin oxide [45]. Our PCA using all strains confirmed De Beer et al.'s [45] observation that SOL-BEi and MAR-ABi are genetically distinct, and SOL-BEi was distinct from the other inbred strains as well. Two strains, MR-VPi and JP-RRi, did cluster nearby each other in the PCA analysis, but were nevertheless distinct along both PC axes 1 and 2, which explained 29.9% and 25.1% of the variance, respectively.

### The inbred strains have diverse and contrasting resistance profiles

To assess whether the five inbred strains vary in acaricide resistance and life histories, we evaluated resistance to 12 acaricides belonging to multiple classes (Fig 2A and S1 Table) as well as the presence of target-site resistance mutations (Fig 2A and S2 Table). The strains were distinguishable in their resistance to most acaricide classes, and several trends stood out. The inbred strains MR-VPi and MAR-ABi retained the high-level, multi-acaricide resistance phenotypes of the progenitor outbred strains from which they were derived (Fig 2A) [25,39]. In contrast,

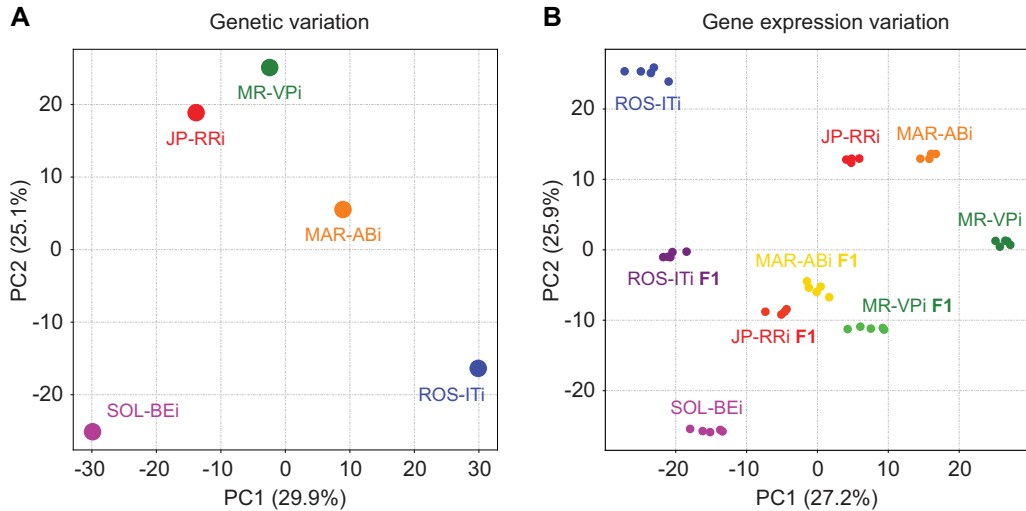

**Fig 1. Genetic and gene expression relatedness among five inbred *T. urticae* strains.** (A) A principal component (PC) analysis based on genetic variation (SNPs) shows relatedness among the five strains, as indicated by points with attending inbred strain names. (B) A PC analysis based on transcriptome profiles shows the similarities among the five strains and the F1 hybrids developed from crosses of four strains (ROS-ITi, JP-RRi, MAR-ABi, and MR-VPi) to a common parent (SOL-BEi). For F1 samples, labeling is based on the parent that varies (i.e., the non-SOL-BEi parent). Transcriptome data was collected from 4–7 day-old adult females, and each dot represents a biological replicate (4–5 replicates were used for each genotype). For both panels, PC1 and PC2 are indicated on the x- and y-axes, respectively (the percent of variance explained by each PC is given in parenthesis).

JP-RRi showed intermediate resistance to most acaricide classes, with modestly higher resistance levels to METI-II compounds. While the resistance profile of SOL-BEi was similar to JP-RRi, it was more resistant to the Na$^+$ channel modulator bifenthrin; although ROS-ITi was highly resistant to dicofol, it showed high susceptibility to most compounds tested (Fig 2A).

To further test for life histories that include exposure to acaricides, we used the genomic sequencing data and variant calls (Fig 1A) to survey target-site resistance mutations across strains (Fig 2A). Target-site mutations that confer resistance to most of the acaricides used in this study have been identified (S2 Table). Both MR-VPi and MAR-ABi have the resistance-conferring H110R mutation in METI-Is' target gene *PSST* (*tetur07g05240*) [41,46], and ROS-ITi and MAR-ABi have the two synergistic resistance mutations in the glutamate-gated chloride channel, G314D and G314E in *GluCl1* (*tetur02g04080*) and *GluCl3* (*tetur10g03090*), respectively, that confer abamectin resistance [47]. The L925M mutation in the voltage gated sodium channel (*VGSC, tetur34g00970*) associated with resistance to pyrethroids in *Varroa destructor* [48–50] was present in all the strains highly resistant to bifenthrin (SOL-BEi, MAR-ABi, and MR-VPi). Further, a recent BSA study identified a candidate target site resistance mutation (V89A) for fenbutatin oxide in the gene encoding subunit c of *mitochondrial ATP-synthase* (*tetur06g03780*) [45] that is present in MAR-ABi and ROS-ITi. However, resistance ratios varied markedly even among strains with the same target-site resistance mutation (e.g., between MR-VPi and MAR-ABi for two of three METI-I compounds, or between MAR-ABi and ROS-ITi for abamectin), and thus other resistance mechanisms must be involved (see Discussion).

## Fitness on different host plants

We also used fecundity as a proxy for evaluating the performance of four of the inbred strains (ROS-ITi, JP-RRi, MAR-ABi and MR-VPi) on five host plant species known to vary greatly in

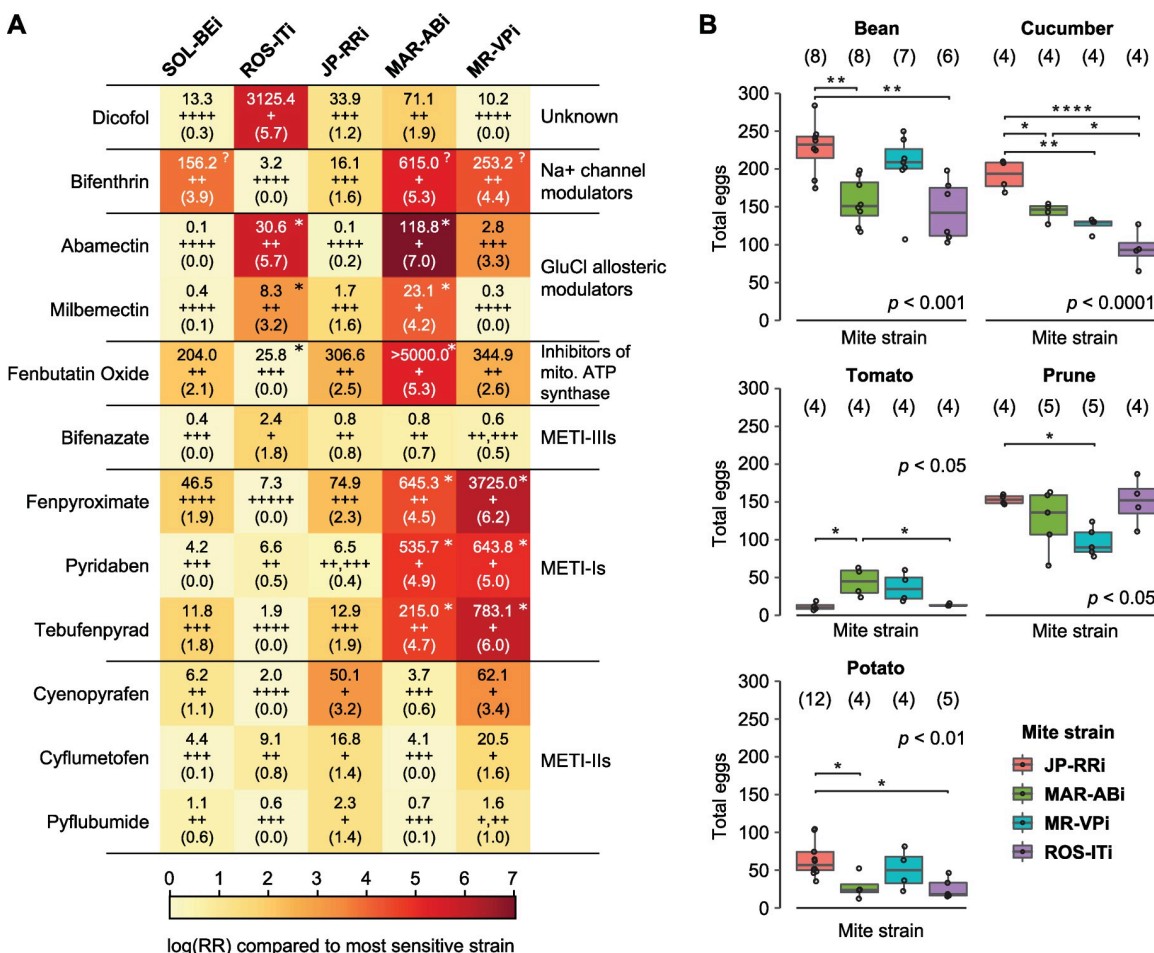

**Fig 2. Resistance phenotypes of inbred strains to twelve compounds and their performance on host plants.** (A) A heatmap showing resistance of five inbred *T. urticae* strains (top) to 12 acaricide compounds (left; IRAC pesticide classes are indicated at right). Log-transformed resistance ratios (RRs) were calculated based on the least resistant strain (scale, bottom). Within each cell, $LC_{50}$ values (mg $L^{-1}$) are shown above the log-transformed RRs (in parentheses). For a given compound, strains for which $LC_{50}$ values are not significantly different from each other have the same number of plus signs (indicated between $LC_{50}$ values and the log-transformed RR values); the number of plus signs negatively correlates with resistance level (i.e., strains with a single plus sign have the highest $LC_{50}$ values for a given acaricide). By strain and compound, an asterisk (top right corner) denotes the presence of established target-site resistance mutations in *T. urticae* (S2 Table); the presence of target site mutations studied in other species, but not validated in *T. urticae*, are denoted by a question mark (bifenthrin only). All phenotypic data is from the current study, excepting that for fenbutatin oxide in strains SOL-BEi and MAR-ABi, which was reported previously by De Beer et al. [45]. (B) The total number of eggs laid by sets of ten mites (each a single replicate) on the adaxial side of bean, cucumber, tomato, prune and potato leaf disks over a 3-day period is as indicated (legend, lower right). The data is displayed using boxplots with an overlay of data points (the number of replicates is given in parentheses). *P*-values (*p*) displayed are from an ANOVA; all significant pairwise differences among strains by host plant are indicated by asterisks ($p < 0.05$: *; $p < 0.01$: **; and $p < 0.0001$: ****, with post-hoc Tukey test to adjust for multiple testing).

their production of specialized compounds and other anti-herbivore defenses [30,51]. These included bean (*Phaseolus vulgaris*), a favorable host widely used for the propagation of laboratory strains of *T. urticae*, as well as tomato (*Solanum lycopersicum*), cucumber (*Cucumis sativus*), prune (*Prunus domestica*) and potato (*Solanum tuberosum*). As shown in Fig 2B, many significant differences were observed among strains on different hosts (ANOVA, with post-hoc t-tests with Tukey corrections for multiple testing, *p*-values < 0.05). For instance, on tomato MAR-ABi had significantly higher fecundity than ROS-ITi and JP-RRi, demonstrating that the strains captured genetic variation impacting fitness on different host plants.

## Variation in gene expression in inbred strains associates imperfectly with genetic distance

To understand the genetic control and inheritance of gene expression variation among the five inbred *T. urticae* strains that vary in acaricide resistance and host performance, we collected RNA-seq data for each of the five inbred strains included in our study, as well as F1 data from crosses of each of ROS-ITi, JP-RRi, MAR-ABi, and MR-VPi to the SOL-BEi strain. Briefly, SOL-BEi was selected as the common parent in the crosses because it was genetically distinct from the other strains (Fig 1A), it was comparatively acaricide sensitive (Fig 2A), and because test crosses revealed that viable F1 progeny were readily produced for each of the four crosses (genetic incompatibilities or partial incompatibilities are not uncommon among *T. urticae* strains [44,52–54]). RNA was collected from 4-7-day old adult females on a single host plant (bean) with either four or five biological replicates for parental strains and resulting F1s (Fig 1B). To assess gene expression variation among genotypes, resulting RNA-seq reads were aligned to the three chromosome London reference genome assembly [42]. To reduce the impact of intra-specific DNA variation among strains on read alignments, genetic variants— identified from genome sequencing and variant prediction (Fig 1A) for each of the five strains —were provided to GSNAP, a variant-aware RNA-seq read mapper (see Methods) [55,56]. A PCA based on transcriptome data revealed that replicates for each strain clustered together and away from other strains, and that F1s had approximately intermediate transcriptomic patterns between each set of parents. While the relative pattern of clustering of parental expression replicates was broadly similar to that observed for genetic variant data, in the PCA based on expression data JP-RRi clustered more closely to MAR-ABi, and further away from MR-VPi, than observed with genetic data (compare Fig 1B to 1A).

## Prominent *trans* control of expression variation among inbred strains

Using a parental/F1 experimental design for each of four pairwise comparisons–ROS-ITi, JP-RRi, MAR-ABi, and MR-VPi compared to SOL-BEi, and the respective F1s –we assessed genetic modes of control for genes with polymorphisms (SNPs) in exons, a requirement for detecting ASE in F1s as needed for genetic mode classification. Among genes that were intact and not copy variable ("intact+single copy" gene sets; see Methods for selection criteria), the number for which assignment of genetic mode of control could be attempted ranged from 8,232 in the JP-RRi × SOL-BEi comparison (43.1% of the total 19,087 annotated protein coding genes) to 8,880 in the MR-VPi × SOL-BEi comparison (46.5%); 4,921 genes (25.8%) could be analyzed in all four of the comparisons (S1 Fig and S3 Table). The *T. urticae* genome annotation includes many small, hypothetical gene predictions [33], a potentially contributing factor to the modest percent of genes included in the analysis (e.g., these sequences may not be functional and therefore intact across strains). In F1 samples for intact+single copy genes for which SNPs were present, we recovered and quantified parent-of-origin reads, and quantified reads from the respective parental replicates at the same sites using the same workflow.

From analyses including the detection of differential gene expression between parental strains and ASE in F1s, we assigned seven modes of genetic control (adjusted $p < 0.1$; see Methods for details). (1) *cis* only (hereafter *cis*), (2) *trans* only (hereafter *trans*), (3) synergism, where *cis* and *trans* act in the same direction on a gene's fold change with respect to a given parent in a pairwise comparison, (4) antagonism, where *cis* and *trans* have opposing effects, (5) compensatory, where there is significant ASE but no differential expression between parents, a result of offsetting effects of *cis*- and *trans*-control on expression (a special case of antagonism), (6) conserved (no evidence of expression variation), and (7) ambiguous (where there are discrepancies in criteria used for assignment to the other modes, and for which

biological interpretation is unclear). In total, genetic mode of control assignments for these categories could be made for between 5,630 (the ROS-ITi × SOL-BEi comparison) to 5,847 (the MAR-ABi × SOL-BEi comparison) of intact+single copy genes (S1 Fig and S4 Table). For 3,381 genes, assignments could be made across all four comparisons. Hereafter, with respect to the differential gene expression analyses used in making assignments, positive $\log_2$ fold change ($\log_2$FC) values indicate upregulation in parent one (P1, either ROS-ITi, JP-RRi, MAR-ABi, or MR-VPi) as compared to parent two (P2, SOL-BEi), with negative $\log_2$FC values reflecting downregulation in P1 compared to P2.

Across the four comparisons 51.9% to 66.4% of genes fell into the *cis*, *trans*, synergism, antagonism, or compensatory categories, with the majority of the remainder in the conserved category (S2 Fig and S4 Table). Among all genes for which a genetic mode of control of *cis*, *trans*, synergism, antagonism, or compensatory could be assigned, those with *trans* control were more numerous by between 2.80- to 3.23-fold as compared to the second most frequent category (*cis*), excepting for the ROS-ITi × SOL-BEi comparison (1.69-fold, Fig 3A–3D; the preponderance of *trans* compared to *cis* control was observed even when a much more restrictive adjusted-*p* value, 0.001, was used for assigning *cis* and *trans* modes, S5 Table). However, with increasing magnitude of $\log_2$FC between parents, the relative number of genes classified as regulated in *trans* decreased markedly relative to those regulated in *cis* (e.g., from ~51.4% to ~16.4% across comparisons in bins of absolute value of parental $\log_2$FC of $< 1$ to $\log_2$FC $>2$, while the fraction of genes with *cis* control increased from ~18.1% to ~32.1%, Fig 4A). Further, as fold change increased, genes in the synergism category increased dramatically, an expected finding (i.e., conditioning on increasing fold change enriches for synergism because the two modes of control, *cis* and *trans*, contribute in the same direction to parental fold change). As shown in Fig 3A–3D, and in Fig 4A, comparatively little difference in these trends was observed between crosses when considering all genes, apart from the relatively high percent of *cis* control in the lower fold change categories in the ROS-ITi × SOL-BEi comparison (Fig 4A).

## Patterns of genetic control for genes associated with xenobiotic resistance and host plant use

Given the phenotypic differences in resistance and host performance among the inbred strains, we examined more specifically modes of genetic control for genes in families implicated in the metabolism, transport, and binding of acaricides and plant defense compounds (detoxification gene set, D, 723 genes), host plant use (host genes, H, 288 genes excluding those with overlap to the D gene set), and a subset of mite peptidases that are potentially involved in overcoming plant produced anti-herbivore protease inhibitors (peptidase gene set, P, 58 genes) (the combined "D+H+P" gene set has 1069 genes; S6 Table). The host gene set consists of genes that were previously associated with adaptation of the reference *T. urticae* strain London to tomato as a new host, most of which currently have an unknown function [24].

Compared to all genes, relatively fewer D+H+P genes fell into the *cis* category in most comparisons, although samples sizes for D+H+P genes were modest, especially following stratification by the absolute value of $\log_2$FC between parents (compare Fig 4A to 4B). Despite the comparatively small sample sizes, however, a greater relative proportion of D+H+P genes were present in categories of higher absolute value of $\log_2$FC between parents than for the background sets of all genes in each comparison (contrast subpanels between Fig 4A and 4B for each of the four comparisons; all *p*-values for chi-square tests between all genes and D+H+P genes were $< 10^{-5}$). Further, among the D+H+P genes with an absolute value $\log_2$FC $> 1$ (between parents, or for ASE in F1s), genes in nearly all detoxification gene families, as well as

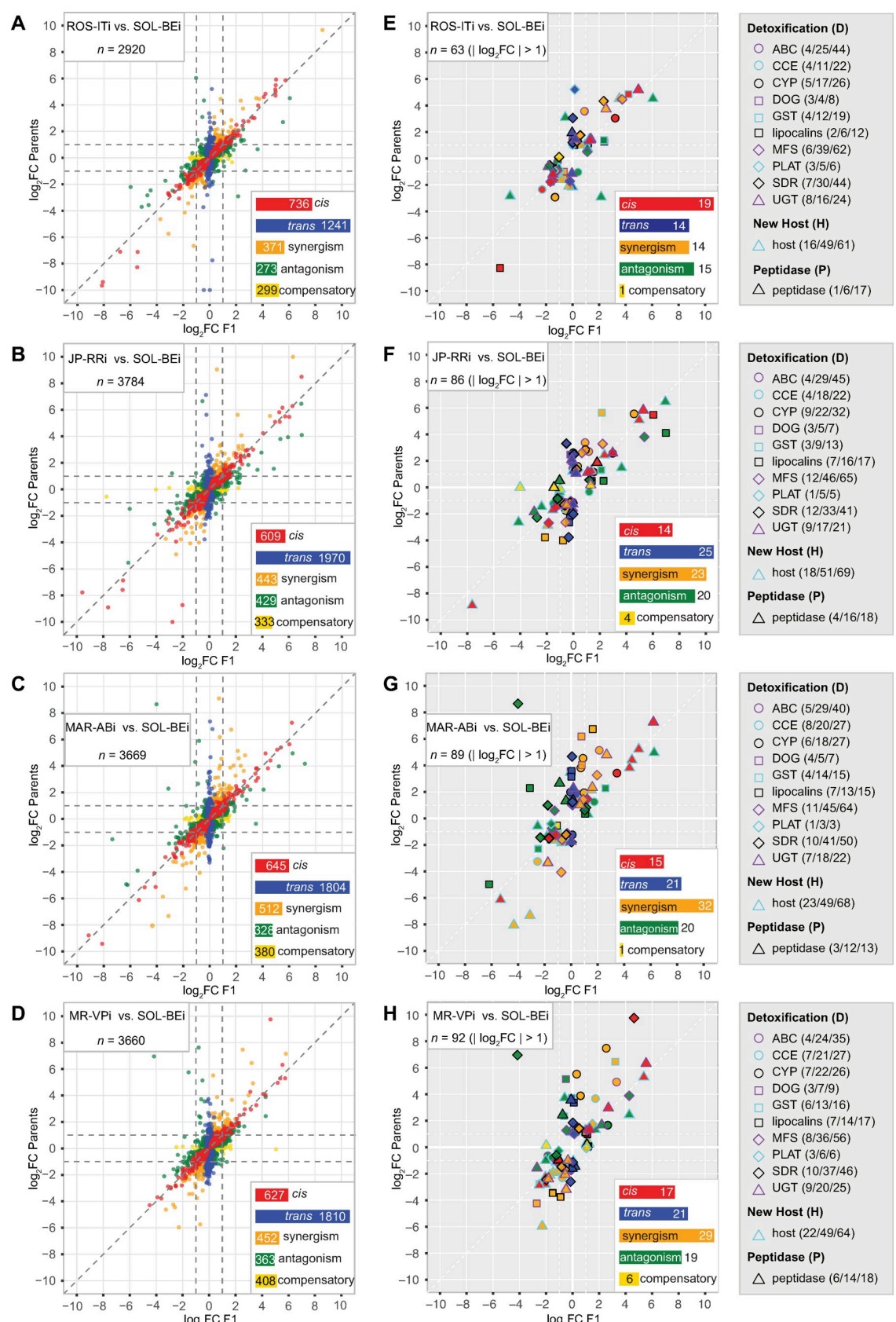

**Fig 3. Regulatory modes associated with gene expression variation in comparisons of parental strains and allele-specific expression in F1s.** (A-D) For the comparisons of ROS-ITi (A), JP-RRi (B), MAR-ABi (C), and MR-VPi (D) to SOL-BEi, scatter plots show log$_2$ fold changes (log$_2$FC) for genes with assigned genetic modes of control between the respective parents (y-axis) versus respective log$_2$FC for allele-specific expression in F1s (x-axis) (genes assigned to conserved or ambiguous modes are not shown; log$_2$FC values falling outside the plotting scale of -10 to 10 were rescaled to these plotting limits). Each point represents a single gene, with color coding by mode of genetic control as indicated (see insets for color codes and respective gene numbers; *n*, total sample size, upper left). (E-H) Respective plots for panels A-D for detoxification/host/peptidase (D+H+P) genes with moderate to large absolute value of log$_2$FC (> 1) along one or more axes. Different symbols and outline colors denote membership in D+H+P gene families or classifications (see legends to the right of each panel). The interiors of plotting symbols are color coded by the regulatory categories as indicated in the insets (*n*, total sample size, upper left). For each D+H+P gene family or classification, the three numbers in parentheses in the legends represent (*i*) the number of genes shown in each panel with *cis*, *trans*, synergism, antagonism, and compensatory mode assignments for which |log$_2$FC| > 1, (*ii*) genes with the same selection criteria except with no log$_2$FC cutoff applied, out of (*iii*) the total number of genes for which any mode of genetic control could be assigned (including conserved and ambiguous; see also S2 Fig). The data upon which this figure is based are provided in S4 Table.

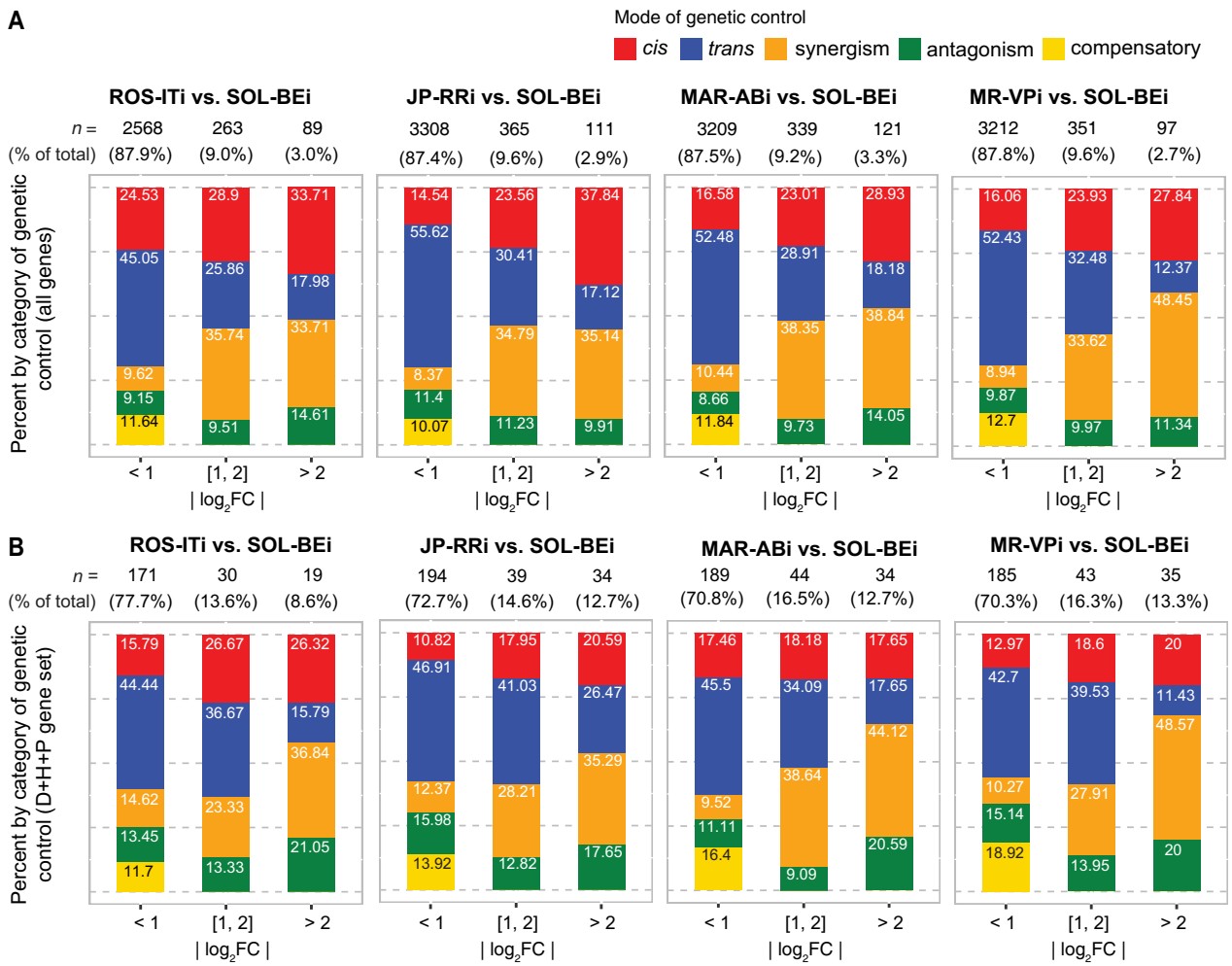

**Fig 4. Composition of genetic control categories stratified by magnitude of parental expression fold change differences.** (A) For all genes for which *cis*, *trans*, synergism, antagonism or compensatory modes of expression control (legend, top right) could be assigned in pairwise strain comparisons, stacked barplots show respective percentages stratified by increasing absolute value of log$_2$ fold change (|log$_2$FC|) bins for parental expression differences (denoted beneath each subpanel; the four genotypic comparisons, as well as respective sample sizes, *n*, are indicated above each subpanel). (B) Same as for panel (A) except for the subset of genes in the detoxification/host/peptidase (D+H+P) gene set. The data upon which this figure is based are provided in S4 Table.

peptidases, were present in each comparison, including a substantial fraction of genes encoding DOGs, lipocalins, and UGTs in one or more comparison (Fig 3E–3H). Overall, host genes followed a broadly similar pattern as for the detoxification gene set. Finally, many genes at the extremes of differential expression magnitude belonged to the D+H+P set (compare Fig 3E–3H to 3A–3D), including both up- and downregulated genes (e.g., for genes with an absolute value log$_2$FC of greater than four, especially for the comparisons of JP-RRi, MAR-ABi, and MR-VPi to SOL-BEi, Fig 3B vs. 3F, Fig 3C vs. 3G, and Fig 3D vs. 3H, respectively).

## CYPs linked to metabolism of known acaricides are among the genes with strongest *trans* regulation

In a complementary approach independent of *a priori* D+H+P gene classification, and to more broadly assess the types of genes and biological functions associated with genetic regulation, we also performed GO term enrichment analyses by modes of genetic control (both molecular function, MF, and biological process, BP; adjusted $p < 0.05$; S7 Table). In analyses of comparisons to SOL-BEi, few GO terms were enriched by genetic mode when using all genes. When stratified by the absolute value of parental fold change, however, several GO terms associated with CYPs and DOGs, like GO:0005506 (MF: iron ion binding), were enriched for *trans*-regulated genes in the JP-RRi comparison where log$_2$FC $\geq 2$. Many of these terms were also enriched in the comparisons involving MAR-ABi and MR-VPi, but for genes controlled instead by synergism. A similar pattern was observed for several other GO terms, including GO:0016758 (MF: transferase activity, transferring hexosyl groups) that comprises UGTs, in either MAR-ABi, MR-VPi, or in both. Further, GO:0008152 (metabolic process) was enriched for synergism in the MR-VPi comparison, with genes contributing to this term's enrichment belonging to the UGT, short-chain dehydrogenase (SDR), and GST families. The modest number of GO terms enriched at log$_2$FC cutoffs of $< 2$ uncover other processes that may vary among strains, but many were observed in only one comparison.

The enrichments observed by genetic mode for pairwise comparisons were, nonetheless, only supported by a small number of genes. Therefore, we also performed a meta-analysis using the upper and lower 5% extremes of log$_2$FC values for significant *cis*- and *trans*-driven control across all four comparisons (genes in the upper and lower tails of the resulting distributions reflect up- or downregulation in strains relative to SOL-BEi; see Methods for additional selection criteria). In this analysis, additional GO terms were enriched, and several trends were apparent (S8 Table). First, only two terms were enriched for extremes of expression variation controlled in *cis*, and only for upregulation relative to SOL-BEi. These terms were GO:0016758 (MF: transferase activity, transferring hexosyl groups) and GO:0008152 (BP: metabolic process), for which more than half of the genes contributing to the enrichments were UGTs and SDRs. Second, a number of GO terms were enriched at extremes of *trans*-driven regulation. For instance, GO terms associated with CYPs and DOGs, like GO:0005506 (MF, iron ion binding), as well as lipocalins (GO:0031409, MF: pigment binding), were enriched at both extremes of fold change relative to SOL-BEi. A similar pattern was observed for GO:0008152, for which genes contributing to enrichment in both directions were dominated by UGTs and SDRs. GO terms uniquely enriched in the upper 5% for *trans*-driven regulation included GO:0008234 (cysteine-type peptidase activity) and GO:0008233 (peptidase activity), which include genes encoding cysteine-type peptidases in several families implicated in digestion of plant tissue [33], as well as GO:0043169 (cation binding) for which enrichment was driven by genes encoding glycoside hydrolases or chitinase (*tetur08g05470*) associated with carbohydrate metabolism. In contrast, among GO terms in the lower 5% of *trans*-driven regulation were GO:0008061 (chitin binding) and GO:0042302 (structural constituent of cuticle).

Despite enrichment of GO terms associated with diverse processes, a striking finding was nonetheless that many genes that contributed to the preponderance of *trans*-enriched GO terms were already in the detoxification gene set. This was especially true for CYPs. Because of this, and because CYPs are important players in detoxification in *T. urticae* and across the animal kingdom [28], we examined more closely the patterns of differential expression and *cis/trans* regulation within this gene family (Fig 5; CYPs for which a genetic mode of control was assigned in at least one comparison are shown). We found that of the eight CYPs enriched in the extreme 5% of *trans*-driven upregulation associated with GO:0005506, five belonged to the CYP392 family (Fig 5, CYP2 clan) previously implicated in acaricide detoxification in *T. urticae* (see Discussion). Compared to SOL-BEi, the most striking upregulation of these genes was observed in the multi-acaricide resistant strains MAR-ABi and MR-VPi, and the least dramatic changes were observed for ROS-ITi, which is comparatively acaricide sensitive. This was particularly striking for the closely related *CYP392A11*, *CYP392A12*, and *CYP392A16* genes, where strong upregulation was explained predominantly in *trans*. However, *trans*-driven upregulation was not universal for CYP392 family members, as both *CYP392B1* and *CYP392B3* exhibited downregulation associated with *trans* effects in MAR-ABi and MR-VPi (and for *CYP392B3*, in other strain contrasts as well). For a few CYP392 genes, *cis* effects did contribute substantially to the overall direction of differential expression between parents (i.e., the upregulation of *CYP392E2* in JP-RRi, or the upregulation of *CYP392D1* in all strains). In a number of instances where synergism or antagonism were observed, the *cis* component of expression variation was minor; nevertheless, *cis*-regulation did contribute synergistically to the very high expression of *CYP392A11*, *CYP392A12* and *CYP392A16* in most strains as compared to SOL-BEi.

In addition to CYP392 genes from the CYP2 clan, *trans*-driven upregulation was also observed for *CYP385A1* and *CYP385B1* from the CYP3 clan and *CYP389B1* from the CYP4 clan, and in general a trend for *trans*-driven upregulation of CYP385 genes was observed, particularly in the comparison with ROS-ITi (although log$_2$FC values were modest as compared to some CYP392 genes). Further evidence that related CYP genes can show divergent expression control in *trans* is provided by three CYP389 genes in the CYP4 clan for which the direction of *trans*-regulation was up for *CYP389B1* (for MAR-ABi), and down for *CYP389C1* (for MAR-ABi) and *CYP389C11* (for JP-RRi, MAR-ABi and MR-VPi).

Although we focused on CYPs, similar patterns were observed for other detoxification genes. For instance, the most striking differences in expression were often explained primarily by *trans* effects, as observed for *DOG1*, *DOG11*, and *DOG13* (in MAR-ABi) (S3 Fig), the lipocalins *tetur24g01030*, *tetur01g05740*, and *tetur01g05730* (in MAR-ABi or MR-VPi) (S4 Fig), or the SDRs *tetur06g05090* (in ROS-ITi and JP-RRi) and *tetur06g04970* (in JP-RRi, MAR-ABi, and MR-VPi) (S5 Fig). As observed for CYP genes, intraspecific regulatory complexity was also readily apparent (S3–S5 Figs and S4 Table). Examples include the lipocalin gene *tetur01g05740*, which was upregulated in both MAR-ABi and JP-RRi as compared to SOL-BEi, with the former explained predominantly by a *trans* effect, while for the latter only a *cis* effect was predicted. In contrast, for the lipocalin gene *tetur04g06010*, *trans* control contributed to large expression differences in MAR-ABi and MR-VPi as compared to SOL-BEi, but in contrasting directions (up in MAR-ABi, down in MR-VPi), reflecting the opposing parental expression differences.

## Dominant inheritance is common for genes controlled in *trans*

In addition to allowing the determination of genetic modes of gene expression control, the parental/F1 experimental design also allowed the assignment of expression inheritance, which

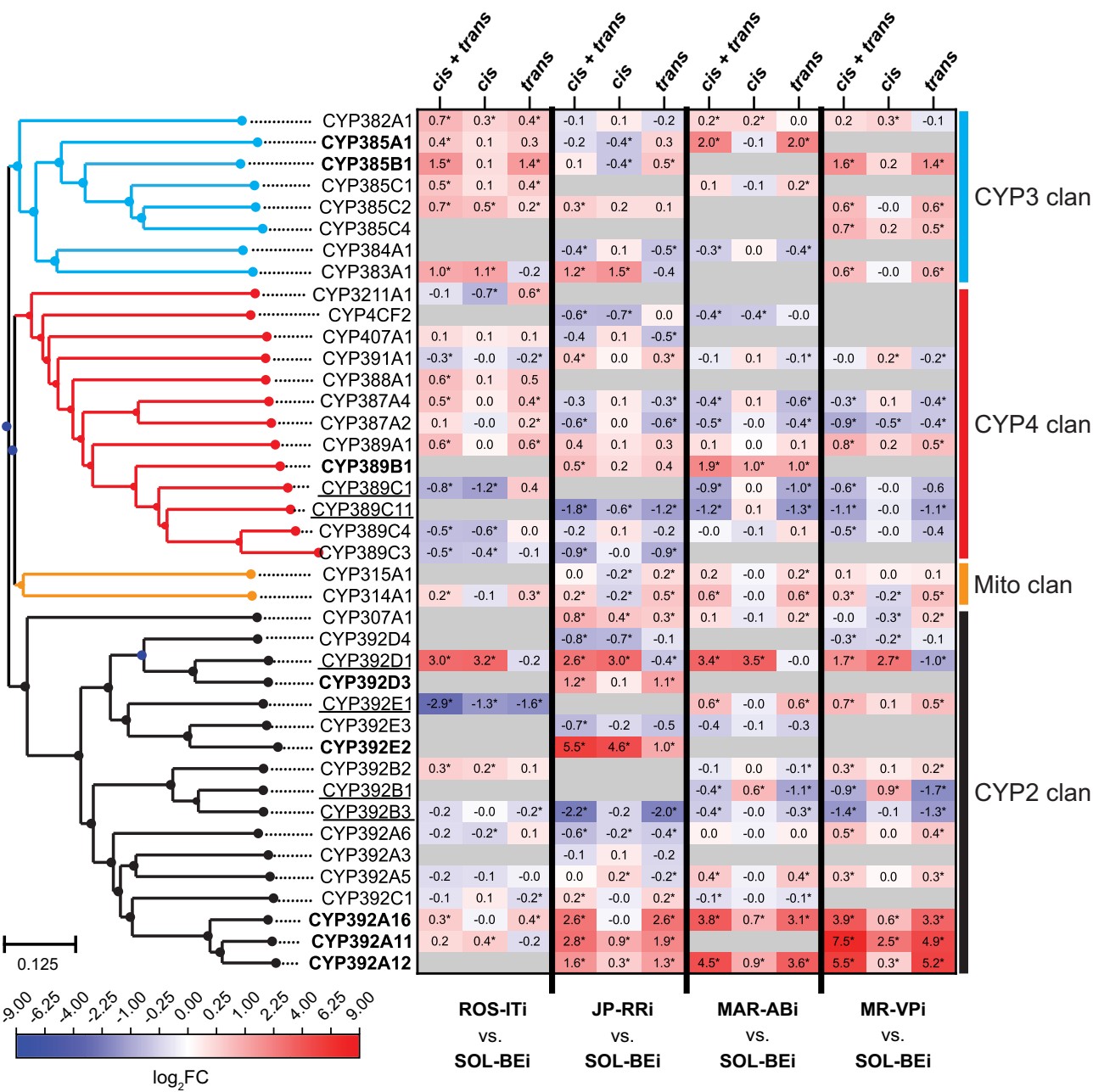

**Fig 5. The contribution of *cis* and *trans* effects for CYP gene expression variation among pairwise strain comparisons.** A heatmap (center right; scale, bottom left) shows log$_2$ fold change (log$_2$FC) values for CYPs between parents (a combination of *cis* and *trans* effects, denoted "*cis* + *trans*") and for ASE in F1s (which correspond to *cis* effects); approximations for the magnitude of *trans* contributions to fold changes were obtained by subtracting *cis* from *cis* + *trans* log$_2$FC values (log$_2$FC values are given internal to each cell; where significant differential expression between parents was observed, or *cis* and *trans* effects were significant, asterisks are used, see Methods and S4 Table). The three groupings are as indicated at the top, and the pairs of strains for the four comparisons are indicated at bottom. Genes with log$_2$FC > 1 attributed to *trans* in at least one of the comparisons are in bold, and genes with log$_2$FC < -1 attributed to *trans* in at least one of the comparisons are underlined. CYP gene names are given to the left, and are ordered based on a neighbor-joining phylogenetic tree (far left). CYP clan membership is indicated at the far right, with clades in the tree color-coded respectively. For inclusion of a CYP in the analysis, a genetic mode of control had to be assigned in at least one comparison. Where information for a gene was not available in a genotypic comparison, cells are colored in gray.

we assessed by using total read counts for all intact+single copy genes (S3 Table; note that the determination of inheritance does not rely on ASE). Gene expression inheritance was assigned using the following categories: additive (when F1 expression equals the intermediate expression of parents), dominant (when F1 expression equals the level of either one of the two parents) and transgressive (when F1 expression falls outside the range of both parents) [12]. Using strict criteria, 8.0% to 15.2% of genes were assigned into one of the three inheritance categories across the four comparisons (adjusted $p < 0.01$, and see Methods; S9 Table). Further, when inheritance modes could be assigned, we also recorded when the expression level for P1 (ROS-ITi, JP-RRi, MAR-ABi, or MR-VPi) was greater than for P2 (SOL-BEi), as well as when P1 was less than P2.

For the comparisons with P2, where there were significant expression differences between parental strains (either P1>P2 or P1<P2), dominant inheritance was most common, accounting for 68.4%, 72.6% and 76.8% of assignments for MR-VPi, MAR-ABi, and JP-RRi, respectively; in contrast, a lower percentage of genes with dominant inheritance was observed in the ROS-ITi comparison (54.6%), for which 43.8% of genes were in the additive category (as compared to 21.6–27.7% for the other three comparisons; Fig 6A and S9 Table). Genes in the transgressive category accounted for a minor proportion of assignments (1.6–3.8%). Additionally, for dominant inheritance there was a bias in the direction of P1 for comparisons between JP-RRi, MAR-ABi, and MR-VPi to SOL-BEi (P2) (2.7- to 5.0-fold more than for dominance in the direction of P2), although this trend was not as striking for the comparison with ROS-ITi (1.3-fold).

We also extended this analysis to specifically examine the D+H+P gene set. Although the resulting number of genes for which inheritance could be assigned was relatively small, broadly similar patterns were observed. Intriguingly, however, a trend for dominance in the direction of the parent with lower expression was observed across each of the four comparisons (1.5- to 4.6-fold more genes than for ones with dominance in the direction of the parental line with the higher parental expression level, Fig 6B). We also assessed, for the set of all genes in the additive and dominant categories, the relationship between absolute $\log_2$FC values for genes with significant differences in expression between parents and the relative proportions of additive and dominant inheritance modes. With increasing parental expression differences, the proportion of genes with additive inheritance increased, while the proportion of dominance decreased (Fig 6C). This pattern mirrored what we observed for modes of genetic control, for which *cis* regulation was proportionally greater for genes in bins of increasing magnitude of parental expression differences (category *cis*, see Fig 4A). Further, we found that the set of loci featuring *cis* genetic control were enriched for additive inheritance, while for *trans* dominance in inheritance was enriched (Fig 6D). A less significant enrichment between *trans* and transgressive inheritance was also observed; in addition, genes controlled by synergism were enriched for additivity, whereas the antagonism category was enriched for transgressive inheritance.

We also examined inheritance patterns for specific detoxification genes with documented or suspected roles in acaricide detoxification in *T. urticae* (see Discussion). Because the inheritance classification which we adapted from Bao et al. [12] only assigned complete dominance or additivity, many instances of putative partial dominance were not assigned, including for D+H+P genes. To describe inheritance patterns more fully for these and other genes, we assigned partial dominance and its direction using a post hoc approach (see Methods and S9 Table). Among CYP genes with strong *trans*-driven upregulation in one or more comparisons (Fig 5, P1>P2), *CYP392A11* and *CYP392A12* (synergism with *trans* most prominent in JP-RRi and MR-VPi, and for the latter also in MAR-ABi) showed partial dominance in the direction of P2, as did *CYP392A16* in the comparison involving JP-RRi (*trans* control). In contrast,

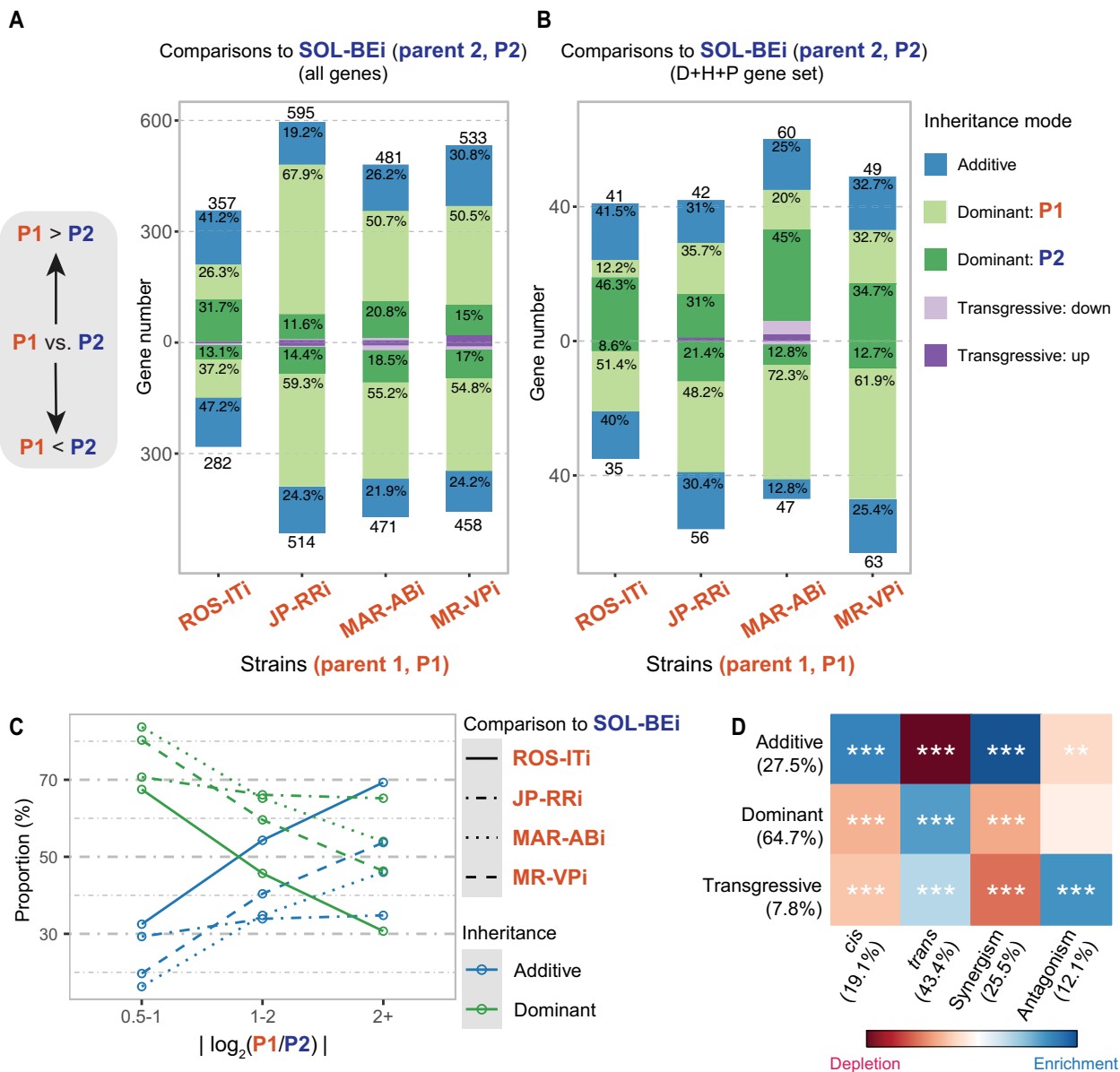

**Fig 6. Inheritance of gene expression and its association with regulatory categories for gene expression variation.** (A-B) Stacked barplots display the composition of each inheritance mode for comparisons of four strains (ROS-ITi, JP-RRi, MAR-ABi, and MR-VPi, indicated as P1, see labels at bottom) to a common parent (SOL-BEi, indicated as P2) for all genes (A) and for those in the detoxification/host/peptidase (D+H+P) gene set (B). The following inheritance classifications were made: (*i*) additive, where that of the F1 equals the average expression of the respective parents, (*ii*) dominant, where that of the F1 equals expression of one parent (the direction of dominance is as indicated in the legend at the far left), and (*iii*) transgressive, for which F1 values are outside the range of the parents (the direction of transgression is defined as either up or down with respect to the parental range). The number of genes (y-axis) for P1>P2 (above 0) and P1<P2 (below 0) are indicated above or below the bars in the plot, with coloring of stacked bars denoting inheritance classifications (see legend, far right; within the stacked bar plots, percentages are given for modes > 5%). In panels A and B, transgressive inheritance for which P1 = P2 (no expression difference between parents) is not indicated. (C) Line plot displaying the relative proportions of additive and dominant inheritance modes (y-axis) stratified by absolute values of $\log_2$ fold change in comparisons between parents (| $\log_2$(P1/P2)|, x-axis). The different line types represent the four comparisons, and different colors denote additive or dominant modes (see legend at right). (D) The association between each combination of mode of genetic control (*cis*, *trans*, synergism, and antagonism) and inheritance mode (additive, dominant, and transgressive). The significance of enrichment (blue) or depletion (red) is denoted using asterisks (chi-square tests of independence followed by Benjamini-Hochberg multiple test corrections; adjusted $p < 0.05$: *; adjusted $p < 0.01$: **; adjusted $p < 0.001$: ***). The data upon which this figure is based are provided in S9 Table.

*CYP392A16* in the comparisons with MAR-ABi and MR-VPi (synergism) exhibited additive inheritance. Among CYP genes with strong *trans*-driven downregulation in comparison to SOL-BEi (P1<P2), additivity was observed for *CYP389C11* in JP-RRi (control by synergism), and dominance in the direction of P1 was observed in MAR-ABi and MR-VPi (control by *trans*). These patterns of variable inheritance among comparisons were observed for a number of other CYPs, as well as for many other D+H+P genes (S9 Table).

## Discussion

With a parental/F1 experimental design we unraveled the genetic control and inheritance of gene expression variation in strains of *T. urticae*, a highly polyphagous mite pest known for its exceptionally rapid evolution of acaricide resistance [19] and host plant adaptation [24,25]. Key findings were that: (1) *trans*-regulatory variation was common; (2) when expression differences of large magnitude were explained by *trans* effects, detoxification genes, including CYPs and DOGs, were enriched; (3) dominance in gene expression inheritance was often observed, and was enriched among genes with *trans*-regulatory control; and (4) dominance or partial dominance was common for many detoxification genes with dramatic expression differences among strains. It should be noted that each of the five inbred strains included in our study harbored at least one target-site resistance mutation (a toxicodynamic mechanism), and some strains like MAR-ABi had multiple target-site resistance mutations and higher resistance to many compounds. However, the relationship between target site mutation numbers and acaricide resistance breadth and levels was incomplete, suggesting an important contribution of acaricide detoxification (metabolic resistance, a toxicokinetic mechanism). Indeed, some target-site mutations in *T. urticae* displayed only moderate phenotypic strength after marker-assisted backcrossing onto a different genetic background [45,57], revealing a requirement for detoxification to attain the very high resistance levels encountered in the field. Further, the strains we used captured variation in host plant use representative of that observed in earlier studies with this species [42,44,58].

### *Trans* regulation is common in *T. urticae*

In all pairwise comparisons, we found that *trans* effects on expression variation were more common than *cis*, even though the latter became more frequent when expression differences between parents were large, as reported in some earlier studies [13,59]. Support for our finding of a higher prevalence of *trans* effects comes as well from inferred modes of inheritance–that is, *trans* effects have been strongly associated with dominance in gene expression ([13], and references therein), and in our study the higher frequency of genes with dominant versus additive inheritance closely mirrored the higher frequency of *trans* versus *cis* effects. For instance, for the ROS-ITi and SOL-BEi comparison, relatively fewer *trans* effects and dominance were detected as compared to the other three (compare Figs 3A–3D and 4A to 6A). The proportion of *trans* versus *cis* effects has been a focus of many studies, although caution is needed in comparing across studies owing to differences in experimental designs and statistical methods. For instance, our study used more biological replicates than most (five for most comparisons), and more replicates facilitate the detection of differential expression of genes with small fold changes (the group of genes that we found to be most enriched for *trans* effects).

Notwithstanding these considerations, our findings are consistent with previous studies in yeast, plants and animals that have used parental/F1 designs with ASE detection to show that while *cis* effects often predominate in interspecific comparisons, over the shorter evolutionary time frame of intraspecific comparisons, *trans* effects are typically, albeit not always, more common ([11], and references therein). Previously, variation in the performance of *T. urticae*

strains on different host plants raised the possibility of host race formation within this species. Despite a striking recent example [44,60], comparatively little evidence exists for wide-spread host-race formation in *T. urticae*, where host plant adaptation can occur in as few as five generations [25], and seasonal or yearly movement of *T. urticae* populations among host plants is likely common. Our finding that *trans* regulation is most often observed in *T. urticae* may reflect this, and contrasts to findings in species like the monkeyflower *Erythranthe guttata* (formerly *Mimulus guttatus*) and the threespine stickleback fish (*Gasterosteus aculeatus*) for which a predominance of intraspecific *cis* variation has been associated with evolutionary divergent subpopulations (i.e., coastal versus inland populations of monkeyflower, and marine versus freshwater populations in the stickleback) [15,61].

## Enrichment for *trans* regulation among detoxification genes

Prior studies have revealed large differences in the expression of detoxification genes between *T. urticae* acaricide-susceptible strains and resistant strains that arise rapidly in response to acaricide application [19], or following adaptation to new host plants [24,25]. Selection on *trans*-acting factors is a potential mechanism underlying rapid intraspecific evolution, as many downstream genes impacting the same biological process can be coordinately changed in expression [62,63]. Our study design allowed us to assess this possibility with *T. urticae* strains, including MAR-ABi and MR-VPi, which were derived from outbred progenitor strains that have experienced recent, strong and recurring acaricide selection [25,41,43,64]. Strikingly, members of the *CYP* and *DOG* families were enriched at the extremes of *trans*-driven expression variation. Recently, several *T. urticae* DOGs including *DOG11* were shown to have broad substrate specificity against aromatic compounds, including plant-derived specialized compounds with roles in plant defense against herbivores [32]. *DOG11* was the most highly over-expressed *DOG* gene in both MAR-ABi and MR-VPi, both of which had comparatively high performance on tomato. RNAi knockdown of *DOG11* in the MR-VPi progenitor strain was recently found to reduce performance on tomato [32], and whether *trans*-mediated upregulation of *DOG11* contributes to the relatively high fitness of these strains on tomato warrants further study. Apart from CYPs and DOGs, it was also striking that specific genes in most other detoxification families, and some associated with host plant use (host use or peptidase gene sets), were also among those with the highest differential expression arising from *trans* control. The relatively low frequency of *trans* effects in ROS-ITi, a strain that like SOL-BEi does not exhibit a high level of acaricide resistance compared to the other strains–particularly the multi-resistant MR-VPi and MAR-ABi strains–is consistent with selection on *trans*-regulatory variation in response to acaricide selection (although a less clear relationship was observed for JP-RRi).

Notwithstanding findings for the *DOG* family, a robust trend observed across detoxification gene families was that only subsets of genes in families were controlled in *trans*, or that the direction of the expression changes (up or down relative to SOL-BEi) within families or among strains often varied. This finding suggests modular regulation of xenobiotic and host plant associated genes in *T. urticae*. Such modular control might facilitate adaptation to specific host plants and acaricides without wholescale activation of xenobiotic pathways, an outcome that likely entails a fitness cost. In fact, in *T. urticae* evidence for costs to heightened metabolic resistance comes from studies involving *cytochrome P450 reductase* (*CPR*), which encodes the electron donor required for CYP activity. In multiple experimental evolution designs using *T. urticae*, the allele frequencies of *CPR* coding sequence and copy number variants hypothesized to modulate or elevate CPR activity, and that coincided with narrowly-defined intervals for acaricide resistance QTLs, decreased rapidly in the absence of continuous acaricide selection [41–43].

While the parental/F1 experimental approach we used to infer genetic control is a powerful design for detecting *trans*-regulatory variation, it cannot identify the loci that are causal. In insects, several transcriptional regulators have been implicated in responses to xenobiotics, including genes in the Cap'n'collar isoform C and hormone receptor 96 (HR96) families (see Amezian et al. [65] for a recent review). Although little is known about regulators of detoxification pathways in *T. urticae*, the genome encodes multiple HR96 family members [41]. Canonical members of this gene family, which encode nuclear hormone receptors with ligand-binding domains (LBDs) and DNA-binding domains (DBDs), have been associated with xenobiotic metabolism in insects such as *Drosophila* [66], and are orthologs of the vertebrate xenosensors PXR and CAR [67]. An HR96 transcription factor (*tetur36g00260*) previously associated with the resistant phenotypes of the MAR-ABi and MR-VPi progenitor strains, and to adaptation to tomato for five generations [25], had *trans*-driven upregulation in all four strains in our study (although antagonistic *cis* effects were also observed in MAR-ABi and JP-RRi). Strikingly, while *D. melanogaster* has one canonical HR96 copy and *T. urticae* has eight, a remarkable 47 genes are present in the *T. urticae* genome that have an HR96-like LDB but that lack a DBD [41]. One of the latter (*tetur06g04270*) was at the peak for a QTL for tebufenpyrad resistance in a cross of a strain derived from the MR-VPi progenitor and a susceptible strain [41]. These gene expression and genetic findings, coupled with the exceptional expansion of HR96-like genes in *T. urticae*, raise the possibility that genetic differences at HR96 loci may contribute to the variation in the expression of subsets of detoxification genes observed in our study.

## Multiple modes of genetic control for CYPs known to metabolize acaricides

We observed large expression differences in CYPs known to metabolize acaricides included in our study. This was especially true for the CYP392A clade that includes *CYP392A11*, *CYP392A12*, and *CYP392A16* (CYP2 clan). Especially in multi-acaricide resistant MAR-ABi and MR-VPi, these genes were outliers among all CYPs in their very high expression as compared to SOL-BEi. CYP392A-clade genes have or are likely to have broad substrate specificity, with CYP392A11 shown to metabolize fenpyroximate and cyenopyrafen [68], and CYP392A16 abamectin [69] and pyflubumide [43]. Additionally, RNAi knockdown of *CYP392A16* reduced abamectin resistance in the progenitor of the MAR-ABi strain [64]. For the CYP392A-clade genes in resistant strains compared to SOL-BEi, upregulation resulted mainly from *trans* effects. Whether the coordinated upregulation of *CYP392A11*, *CYP392A12*, and *CYP392A16* in resistant strains originates from variation in a shared upstream regulator, and whether such a regulator might also control genes in other detoxification families, is an outstanding question.

Finally, although *trans* effects were typically largest in magnitude among CYPs, *cis* variation was also observed, including for CYP392A-clade genes. In fact, in a prior study promoter variation among two strains, including the MAR-ABi progenitor strain MAR-AB, was demonstrated for *CYP392A16*, with higher expression observed from the MAR-AB promoter than from the promoter of an acaricide-susceptible strain [64]. Adaptive roles for *cis* variants in the control of known detoxification genes in insects have also been established or suggested by a number of studies [70,71]. It was striking that synergism was often observed where both *trans* and *cis* control of CYP392A-clade genes was inferred. However, among *CYP392A11*, *CYP392A12*, and *CYP392A16*, a moderately large contribution of a *cis* effect was only observed for *CYP392A11* in strain MR-VPi. The extent to which synergism underpins extremes of expression differences in detoxification gene expression is incompletely understood. Nevertheless, our findings suggest that the evolution of high-level resistance phenotypes in response to

acaricide exposure may involve selection on both *cis* and *trans* variants acting in a reinforcing manner. In particular, synergism underlying the exceptionally high expression of detoxification genes with broad substrate specificities, like *CYP392A11*, may have outsized importance.

## Dominance or partial dominance is a common inheritance pattern among detoxification genes

The ability of advantageous alleles to respond to selection depends on multiple factors, including the degree of dominance. In diploid individuals, dominant mutations are unmasked to selection in the heterozygous state, and hence selection on dominant variants can occur rapidly even when allele frequencies are low (in populations, low frequency alleles are found predominantly in the heterozygous state). Given the rapid evolution of acaricide resistance in *T. urticae*, an expectation might be that the high expression of genes such as those in the CYP392A-clade would result from dominance in the direction of resistant strains like MAR-ABi and MR-VPi. While a trend toward dominance of overall gene expression in the direction of these strains was observed, the direction of dominance for D+H+P genes was instead biased toward the parent with lower expression, even if substantial heterogeneity was observed for specific genes. The significance of this trend, as well as if this could be an effect of the single common parent (SOL-BEi) used in our experimental design, is currently not clear. More speculatively, it might reflect the action of purifying selection in *T. urticae* populations on variants that in the heterozygous state result in strong upregulation of detoxification and host-associated genes, but that may also have negative pleiotropic effects (i.e., fitness costs of elevated detoxification in the absence of continuous selection by acaricides or challenging host plants). Regardless, for many detoxification and host plant genes, including *CYP392A11*, *CYP392A12*, and *CYP392A16* as well as many *DOGs* that also had very high *trans*-driven expression in the multi-acaricide resistant strains, partial dominance or additivity was observed. The finding of dominance, or the otherwise intermediate expression of many detoxification genes in F1s as compared to parents (i.e., partial dominance or additivity), suggests that genetic variants underlying expression of xenobiotic resistance or host plant use associated genes should be visible to selection in female *T. urticae* mites (the diploid sex used in our study). This is consistent with the findings of a number of studies that have revealed that phenotypic resistance against pesticides often exhibits incomplete dominance contributed by multiple loci [72–75]. For spider mites, however, an additional factor that likely impacts rapid adaptation is that males are haploid, and in haploid individuals dominance relationships are not relevant. Extrapolation of our findings about gene expression in *T. urticae* females to males will nonetheless require further study.

## Conclusion

Among *T. urticae* strains we found that large changes in the expression of genes known or suspected to be important for detoxification or host plant use were often underpinned by variation in *trans*-regulatory environments. This was especially striking for genes encoding CYPs documented to detoxify multiple acaricides, and DOGs that can metabolize a range of plant-derived aromatic compounds. The extent to which high expression of genes encoding detoxification enzymes with broad substrate specificities can serve as biomarkers to assess the resistance potential of *T. urticae* strains warrants future investigation. More generally, our findings position *T. urticae* as an attractive system for future studies to characterize the molecular basis of the control of gene expression variation in xenobiotic and host plant use pathways.

## Methods

### Mite strains

Inbred strains used for genetic and gene expression analyses originated from five *T. urticae* strains that we maintained on potted bean plants or detached bean leaves on wet cotton (*Phaseolus vulgaris* var. 'Prelude'): ROS-IT, MAR-AB, MR-VP, JP-RR, and SOL-BE. Briefly, SOL-BE was collected on potato in Belgium in 2018, and ROS-IT was collected from roses in Italy in 2017. The MAR-AB strain originated from a heavily sprayed greenhouse near Athens (Greece) in 2009, and is resistant to abamectin, bifenthrin, clofentezine, hexythiazox, fenbutatin oxide and pyridaben [25]. The MR-VP strain originates from a lab stock originally collected in September 2005 from bean plants in a greenhouse in Brussels, Belgium, and was selected for its extraordinary levels of resistance to mitochondrial electron transport inhibitors of complex I (METI-Is) [39]. The laboratory strain JP-RR was selected for its resistance to electron transport complex II (METI-II) inhibitors (cyenopyrafen, cyflumetofen, and pyflubumide) from its parental strain JP-R, as described by Khalighi et al. [40]. To avoid the confounding effect of heterozygosity on downstream studies of the control of gene regulation, the strains were inbred via mother-son mating performed on detached bean leaves as described by Bryon et al. [38] for 7 to 9 generations to produce the respective inbred strains (postfixed with the "i" for inbred designation; ROS-ITi, MAR-ABi, MR-VPi, JP-RRi, and SOL-BEi). After the final round of inbreeding, the strains were expanded and maintained on potted bean plants to produce sufficient mites for downstream analyses (no acaricide selection was applied either during or after inbreeding). All strains were maintained at 25°C (±0.5°C), 60% relative humidity, and 16:8 h light:dark photoperiod.

### Bioassays

Acaricide resistance levels of all inbred strains were determined in bioassays as previously described by Van Pottelberge et al. [39]. Serial dilutions of acaricides and water (negative control) were tested in four replicates of 20–30 adult female mites on 9 cm$^2$ square bean leaf disks on wet cotton. Using a Cornelis spray tower, 800 µl of fluid was sprayed on each leaf-disk at a pressure of 1 bar (1.5 mg fluid deposition per cm$^2$). The acaricides used and the corresponding IRAC Mode of Action (MoA) and time of mortality assessment are listed in S1 Table. Dose-response curves, LC$_{50}$-values, and their 95% confidence intervals were calculated using POLO-plus software (version 2.0, LeOra Software); after earlier work, significant differences were assessed when 95% confidence intervals did not overlap [76].

### Host performance assays

The fitness of inbred strains ROS-ITi, JP-RRi, MAR-ABi, and MR-VPi was assessed on five different hosts: bean (*Phaseolus vulgaris* var. 'Prelude'), tomato (*Solanum lycopersum* var. 'Moneymaker'), potato (*Solanum tuberosum* var. 'Bintje'), cucumber (*Cucumis sativus* var. 'Ventura') and prune (*Prunus domestica* var. 'Brompton'). This was evaluated by placing leaf discs of the respective host plant (4 cm$^2$) on wet cotton wool with the adaxial side up. Discs were infested with ten female adults (4–5 days old) per disc with at least four replicates per strain and per host. After three days, the total number of eggs on each leaf disc was recorded. Statistical differences among strains by host plant were assessed by analysis of variance (ANOVA); where significant differences were observed, post-hoc Tukey tests were applied. Analyses were performed in R (version 4.0.4) [77].

### DNA extractions and sequencing

For each of the parental inbred strains, DNA was isolated from a pool of 400 adult female mites by phenol-chloroform extraction as described in Villacis-Perez et al. [44]. DNA concentration and purity were measured using a spectrophotometer (DeNovix, USA). Integrity of the DNA samples was checked via gel electrophoresis (2% agarose gel; 30 min; 100V). From the resulting DNA samples of the MAR-ABi, MR-VPi, JP-RRi, and SOL-BEi strains, Illumina DNA-seq libraries were constructed using the Illumina TruSeq Nano DNA library prep kit, and subsequently sequenced with the Illumina HiSeq2500 technology (paired-end reads of 125 bp) at the Huntsman Cancer Institute of the University of Utah (Salt Lake City, Utah, USA). From the DNA extracted for ROS-ITi, an Illumina library was constructed using the Illumina NEBNext Ultra II DNA library prep kit and sequenced with the Illumina HiSeq3000 technology (paired-end reads of 150 bp) at NXTGNT (Ghent, Belgium). Resulting coverage depths ranged from approximately 98- (MAR-ABi) to 430-fold (ROS-ITi).

### DNA read alignments and PCA

Genomic reads from each of the five inbred strains were aligned to the *T. urticae* London reference genome [42] using BWA (version 0.7.17-r1188) [78]. Sorting and indexing of the resulting BAM files were performed with SAMtools (version 1.9) [79], and duplicate reads were marked with Picard (version 2.18.11-SNAPSHOT) (http://broadinstitute.github.io/picard/). Genetic variants across all strains were predicted with the Genomic Analysis Toolkit (GATK) following the best practices workflow (version 4.0.7.0) [80,81]. To assess potential acaricide resistance target-site changes, effects of predicted variants on amino acid sequences were assessed using SNPeff (version 4.3L) [82] based on coding exons (CDS) from the June 2016 ORCAE GFF annotation [83]. To describe relatedness among strains, a principal component analysis (PCA) was performed after filtering the resultant variant call format (VCF) file with additional criteria to further eliminate aberrant predictions. The filtering criteria were adapted from previous studies [42]. Briefly, each SNP included in the PCA had to have the quality score normalized by allele depth (QD in the VCF file) of at least 2, mean mapping quality score (MQ) of at least 50, and strand odds ratio (SOR) below 3. Mapping quality rank sum (MQRankSum) and rank sum for relative positioning of alleles in reads (ReadPosRankSumTest) both had to be at least −8 when present. Finally, coverage (as indicated by the total depth per allele per sample [AD] field in the VCF file) had to be within 25% and 150% of the sample's genome-wide mean SNP read coverage (to restrict the analysis to variants in putatively single copy regions). The PCA was carried out in R (version 3.5.2) [77] using SNPRelate (version 0.16.1) [84] with the "autosome.only = FALSE" option. The PCA was performed on the GDS file converted from the VCF using gdsfmt (version 0.18.1) [84,85] with the option "method = copy.num.of.ref" (which includes all SNPs). Further, we applied the approach of Villacis-Perez et al. [44] to assess if the inbred strains were in fact genetically isogenic; briefly, by strain and SNP site, we used alternative and reference allele count data in VCF files to assess the fraction of SNPs supported by only one allele (i.e., the fraction that were fixed, an expectation for isogenic strains).

### Genetic crosses for genetic mode of expression control determination

To investigate expression variation between strains and its genetic mode of control and inheritance, we adapted a parental/F1 design that has been used previously [8,11–14]. To do this, we crossed 40 virgin females of strains ROS-ITi, JP-RRi, MAR-ABi, and MR-VPi to 40 males of strain SOL-BEi to generate four sets of F1s; in parallel, we produced matching progeny for each parental strain. The crosses to generate F1 and parental strain females were performed

with 5-fold biological replication, except for MAR-ABi, JP-RRi, and the JP-RRi × SOL-BEi F1s, for which four biological replicates were obtained. For each replicate (F1 or parental), 100–120 adult female mites at four days of age were collected for RNA extraction, except for the JP-RRi x SOL-BEi F1s for which females of 4–7 days old were collected. Immediately after mite collection, samples were frozen in dry ice until storage at -80˚C.

## RNA extractions and sequencing

For each frozen sample of 100–120 adult female mites (a parental or F1 replicate) total RNA was extracted using the RNeasy Plus Mini Kit (Qiagen, Belgium) according to the manufacturer's Quick-Start Protocol. RNA concentration and purity were measured using a DeNovix spectrophotometer (DeNovix, USA). The integrity of the RNA-sample was checked via gel electrophoresis (1% agarose gel; 30 min; 100 V). Illumina libraries were constructed using the Illumina TruSeq stranded mRNA library prep kit and sequenced on the Illumina HiSeq3000 platform (paired-end reads of 150 bp). Library construction and sequencing was conducted at NXTGNT (Ghent, Belgium).

## RNA-seq read alignments and PCA

RNA-seq reads from parents and F1 samples were aligned to the London reference genome using GSNAP (version 2018-07-04) [55,56], an aligner that incorporates SNP information to reduce reference bias in the alignment of reads derived from non-reference samples. The alignments were performed with known splice sites supplied to GSNAP using the GFF annotation of Wybouw et al. [42] with novel splice discovery enabled and with the options "—locals-plicedist = 50000—novelend-splicedist = 50000—pairmax-rna = 50000—gmap-mode = all". Variant information was supplied to GSNAP in a sample-dependent manner. Briefly, pairwise VCF files were generated and supplied to GSNAP for the respective alignments. For instance, for alignments of reads from F1s from the ROS-ITi × SOL-BEi cross, a VCF file was supplied to GSNAP that contained only the SNP variants identified for the ROS-ITi and SOL-BEi strains as compared to the London reference genome (see section "DNA read alignments and PCA"). Read counts per gene per genotype and biological replicate were then assessed with HTSeq (version 0.11.2) [86]. To assess global patterns of gene expression, as well as variation among replicates for all parental strain and F1 samples, a PCA was performed on the resulting expression data using DESeq2 (version 1.22.2) [87].

## Selection of putatively intact and single copy gene sets

For pairwise strain expression comparisons to detect ASE (i.e., for each set of two parent strains and the respective F1), we first identified genes that were putatively intact and single copy in each pair of parents. To select the "intact+single copy" gene sets, we used both genomic read and RNA-seq read data. With aligned genomic reads on a per strain basis, we detected putative copy number variation using read coverage. To do this, median per-base DNA read coverage was determined across the three *T. urticae* chromosomes. Briefly, pysam's "count coverage" function was used with default settings (pysam version 0.15; https://pysam.readthedocs.io/en/latest/api.html), and the median of all non-zero values was recovered. In addition, the coverage for each CDS exon of a gene was assessed using the same approach with reads that mapped in a proper pair and with a MQ score of 40 (the maximum quality score). Genes that had a coverage below 5% or above 150% of the genome-wide median within any CDS exon were considered to be copy variable in the respective strain. In addition, GATK variant predictions (single nucleotide and small indel differences) were assessed for impacts on genes by substituting them into the London reference sequence on a per strain basis. The

respective CDS sequences for each gene were then extracted and joined to determine the largest open reading frame (ORF) accounting for the strand on which the gene was annotated ("+" or "-"in the GFF annotation file). Biopython (version 1.74) was then used for translations [88]. To account for variation at gene ends that may be neutral, such as nearby alternative start or stop codons, sequences 12 bp upstream of the first CDS exon and 12 bp downstream of the last CDS exon were included in ORF predictions.

In order for a gene to be considered for inclusion in downstream ASE analyses involving a pair of strains based on genomic read data, the following criteria had to be met: (1) the gene had to be putatively single copy in both strains as assessed from DNA read coverage, (2) the longest ORF of the gene had to contain a putative start and stop codon in both strains, and (3) there had to be an amino acid sequence match of at least 90% for the predicted protein sequences between the two strains. For assessing the latter, global alignments were performed using the pairwise2 module in Biopython; identical amino acids were given a score of 1, and the sum of scores was divided by the alignment length (the longer of the two sequences used in each pairwise comparison) to determine percent identity.

Additionally, to be considered in an ASE analysis between a pair of parents, a gene had to pass a final filter based on RNA-seq read alignment data. Briefly, apparent heterozygosity in aligned RNA-seq reads, if observed in inbred strains, may result from cross-mapping of similar, duplicate sequences to a single sequence in the reference genome used for alignment. To detect this, for each inbred strain we combined RNA-seq reads from all replicates (e.g., from all BAM files for a given parental genotype) and selected reads in proper pairs with MQ values of 40. For each position within a CDS, we required that the minor allele frequency could be at most 0.1 (in assessing base calls in reads by position, a minimum base quality score threshold of 30 was used). If a gene failed this criterium in either of the two parents, it was not included in the respective intact+single copy gene set.

## Assignment of parent of origin to aligned RNA-seq reads

ASE analyses based on RNA-seq alignments require that individual reads are assigned to a parent of origin based on variants in transcribed regions (reads that align to identical sequences between the two parent strains are uninformative). For F1 samples, we developed an in-house Python program that employed pysam (version 0.15) to distinguish reads derived from one or the other parent for a respective F1 sample. Briefly, SNPs that distinguished the two parents of a given F1 were first recovered from the respective pairwise VCF files (see section "RNA-seq read alignments and PCA"; an MQ value of 40 or more was applied for SNP selection). F1 RNA-seq reads that were mapped in a proper pair with read-level MQ values of 40 (the maximum values assigned with GSNAP) were then recovered and written to parent of origin BAM files if reads contained bases at SNP sites specific to one of the two parents of the F1 (reads for which the parent of origin could not be assigned were discarded). The same workflow was also applied to RNA-seq samples of the respective parents to generate matching read alignment data (that is, only parental reads that overlapped the same informative SNP sites used in the F1 analyses were included). Following parent of origin read assignment, reads within CDS exons were counted from the resulting BAM files with HTSeq.

## Assignment of mode of genetic control for gene expression variation

To assess modes of genetic control, we first performed differential gene expression analyses between parents of an F1 (to detect combined effects of *cis* and *trans* control) and between parent-of-origin reads for the respective F1s (to detect *cis* effects reflected by ASE) with DESeq2 (adjusted *p*-value < 0.1). Lowly expressed genes were excluded from subsequent analyses (base

mean value < 20). Second, we applied Fisher's exact test to DESeq2 normalized allele count ratios of F1 versus normalized count ratios of respective parental strains for genes in the "intact+single copy" gene sets by pairwise comparison (if these ratios differ, a *trans* effect is indicated [14]; Benjamini-Hochberg correction, adjusted-$p$ < 0.1). For performing the ratio tests between the F1s and parents, the respective normalized expression data across replicates was summed, and resulting values were weighted by the number of replicates if different numbers of replicates were available in a given pairwise comparison. Associated with these analyses, the fold changes between parents, as well as those for ASE in F1s (*cis* effects), were from estimates from the DESeq2 analyses ($log_2FC$ values) that were shrunk using ashr (version 2.2–32) [89] to prevent genes with lower expression from confounding the results. To provide approximations of the magnitude of contributions of *trans* effects to genic fold changes, $log_2FC$ values for the analyses of F1s were subtracted from the parental $log_2FC$ values.

Finally, with these results for the parents (combined *cis* and *trans*), F1s (ASE, *cis*), and ratios (*trans*), we classified genes into seven control modes by adapting established criteria [13,14]: (1) *cis* (parents and F1s significant, ratios not); (2) *trans* (parents and ratios significant, F1s not); (3) synergism (parents, F1s, and ratios significant, and hence *cis* and *trans* assigned, and with the effects on expression of *cis* and *trans* in the same direction with respect to a given parent), (4) antagonism (parents, F1s, and ratios significant, and hence *cis* and *trans* assigned, and with the effects on expression of *cis* and *trans* in the opposing directions), (5) compensatory (parents not significant, but F1s and ratios significant, reflecting *cis* and *trans* effects that offset), (6) conserved (parents, F1s, and ratios all not significant), and (7) ambiguous (any pattern incongruent with the above classifications).

## Inheritance of gene expression variation

By comparing expression of genes between the parental strains and that of respective hybrid F1 offspring, the inheritance mode for expression of individual genes was inferred. For this purpose, we adapted the methodology from Bao et al. [12]. The inheritance modes were classified into the following categories: "additive" (expression in F1s is equally between that of the two parental strains), "dominant" (F1 expression is equal to that of either the maternal or paternal strain), and "transgressive" (F1 expression falling outside the parental range). Note that while additivity and dominance are observed when there are significant expression differences between parental strains, transgressive inheritance can be observed even if there is no expression divergence between parents. For the inheritance mode classification, we performed four rounds of differential gene expression analyses using DESeq2 (version 1.28). Input for the analyses was total gene read counts for all genes (all replicates were used for every genotype, parental or F1); for the following, P1 represents the varying parents (ROS-ITi, JP-RRi, MAR-ABi, and MR-VPi) and P2 represents the SOL-BEi strain. The first differential expression analysis was performed using the parents, and significantly differentially expressed genes between parents (P1≠P2) were identified with the following criteria: (1) $p$-adjusted < 0.01, (2) absolute $log_2FC$ > 0.5, and (3) baseMean > 15 (to remove lowly expressed genes from the analysis). The second and third rounds of analyses were performed by comparing F1 expression to the respective two parental strains using a significance cutoff of $p$-adjusted < 0.01. The fourth analysis was performed by comparing F1 expression to the midpoint of parental strains ($p$-adjusted < 0.01; see Bao et al. [12] for methods). In the case of P1≠P2, if F1 = P1 or F1 = P2 only, genes were classified into the dominant category in the direction of one or the other parent, and for F1 = (P1+P2)/2 only, genes were classified into the additive inheritance category. The criteria for transgressive inheritance were (1) F1>P1 and F1>P2, or (2) F1<P1 and F1<P2.

The strict criteria we used to assign genes into pure dominant, additive, and transgressive categories overlooked a large number of genes showing partial dominance when $P_1 \neq P_2$. To more fully describe inheritance, we used the formula of Stone [90] to assign partial dominance and its direction:

$$H = \frac{2F1 - P1 - P2}{P1 - P2}$$

where F1, P1 and P2 represent respective mean expression values of each genotype (after normalization by DESeq2). When $0 < H < 1$, incomplete dominant in the direction of P1 was assigned; when $-1 < H < 0$, incomplete dominant in the direction of P2 was assigned.

Association analyses between inheritance modes (excluding partial dominance categories) and modes of gene regulatory control were based on Pearson's Chi-squared test using the chisq.test function in R (version 4.0) [77] with data across all four comparisons for the subset of genes where both inheritance modes and gene regulatory modes were assigned.

## Detoxification and host plant associated genes

"Detoxification genes" were defined as genes belonging to gene families involved in detoxification and/or host plant adaptation, although it should be noted that not every gene in these families plays a role in the detoxification process. For ATP-binding cassette transporters (ABCs), carboxyl/cholinesterases (CCEs), glutathione S-transferases (GSTs), intradiol ring-cleavage dioxygenases (DOGs), and UDP-glycosyltransferases (UGTs) gene family information was obtained directly from the *T. urticae* three-chromosome gene information file [42]. Since the release of the 2016 annotation, Snoeck et al. [37] reannotated short-chain dehydrogenases (SDRs) and PLAT-domain proteins in *T. urticae* and gene identifiers for those families were obtained from "S7 and S8 Tables" in their manuscript (only complete sequences were used). In addition, genes belonging to the cytochrome P450 (CYP) family and the major facilitator superfamily (MFS) were identified using InterProScan (version 5.32–71.0) [91], which was run on the peptides specified in the June 2016 ORCAE annotation [83], by extracting genes annotated as "IPR036396" ("Cytochrome P450 superfamily") and "IPR011701" ("Major facilitator superfamily", Pfam category identified as upregulated upon host acclimation in Snoeck et al. [37]), respectively. The lipocalin gene set came from "S2 Table" in Dermauw et al. [25].

The "core" genes that were highly and consistently differentially expressed upon host transfer in Wybouw et al. [24] were also included and referred to as "host" genes (288 after excluding genes that overlapped with the detoxification gene set). For putative digestive proteases, we focused on the large C1A (papains, cathepsin L) subfamily of cysteine peptidases as listed in column one of Grbić' et al.'s [33] supplementary table "Table S6.1.9. Cysteine peptidase genes in *T. urticae*" (note that the *tetur13g02490* identifier was replaced by *tetur13g02500* in later genome annotations). See S6 Table for resulting family and host classifications for genes.

## Gene ontology (GO) enrichment analysis

GO annotations were retrieved from the ORCAE database for *T. urticae* (annotation version 20190115) and enrichment analyses were performed with the enricher function of the cluster-Profiler package (version 4.2) in R (version 4.1) [92] with arguments "pvalueCutoff = 0.05, pAdjustMethod = BH." For each pairwise strain comparison, and conditioned on the parental strain fold change cutoffs given in S7 Table, enrichment analyses were performed for all assigned categories of genetic control modes excepting ambiguous (the background gene sets used for these analyses consisted of the genes with regulatory assignments by comparison,

S4 Table). In a separate analysis, we first constructed distributions for log$_2$FC values for significant *cis* and *trans* effects by pairwise strain comparisons (significance of *cis* and *trans* effects was after Methods section "Assignment of mode of genetic control for gene expression variation"); where a gene was controlled by combinations of *cis* and *trans* (e.g., categories synergism, antagonism, or compensatory), the log$_2$FC approximations for *cis* and *trans* effects (S4 Table) were considered independently in constructing the distributions. Second, from the resulting *cis* and *trans* distributions, we combined the respective upper and lower 5% tails across the four comparisons by taking the union of genes in the respective tails (note that for this analysis, the upper tails for *cis* and *trans* represent extremes of upregulation across ROS-ITi, JP-RRi, MAR-ABi, and MR-VPi as compared to SOL-BEi, while the lower tails reflect downregulation relative to SOL-BEi). Finally, we performed the GO enrichment analyses on the combined *cis* upper and lower tails and the combined *trans* upper and lower tails using the set of all genes for which genetic mode assignments could be made across all pairwise comparisons as the background gene set (column "Gene ID" in S4 Table). Significantly enriched GO terms supported by only a single gene were excluded from subsequent analyses.

### Phylogenetic constructions of detoxification gene families

Neighbor-joining trees were constructed for the CYP, DOG, lipocalin, and SDR detoxification gene families. Amino acid sequences for the respective families were extracted from the June 2016 annotation [83] and aligned using MUSCLE (version 3.8.1551) [93]. The scikit-bio Python package (version 0.5.4) (https://github.com/biocore/scikit-bio) was used to construct neighbor-joining trees based on a distance matrix between the amino acid sequences. The trees were visualized alongside heatmaps displaying differential expression magnitude (log$_2$FC) between parental strains, as well as expression explained by *cis* and *trans* effects, using the Python package ete3 (version 3.1.1) [94] and matplotlib [95]. For display, a gene was included in a given pairwise comparison if a genetic mode of control could be assessed.

### Supporting information

**S1 Fig. Gene numbers for inclusion in assessing genetic mode of control of expression variation.** (A) A Venn diagram shows the number of intact+single copy genes available for potential assessment of genetic modes of control in comparisons of strains ROS-ITi, JP-RRi, MAR-ABi, and MR-VPi to SOL-BEi, and the respective overlaps (see S3 Table). (B) A Venn diagram after panel (A) but indicating the number of genes in each comparison, and the overlap among comparisons, where an assignment of genetic mode of control could be made (see Methods and S4 Table).
(EPS)

**S2 Fig. The composition of genetic mode of gene expression control in pairwise strain comparisons by gene group.** For pairwise comparisons of ROS-ITi (A), JP-RRi (B), MAR-ABi (C), and MR-VPi (D) to SOL-BEi, the number of genes assigned into one of seven genetic control categories is indicated above the stacked barplots (categories *cis*, *trans*, synergism, antagonism, compensatory, conserved, and ambiguous are as indicated by color in the stacks, see the legend at top). In each of panels A-D, classifications are shown for all genes in the left subpanel, and at right for genes in the detoxification/host/peptidase (D+H+P) gene set sorted by gene family or group. The data upon which this figure is based are provided in S4 Table.
(EPS)

**S3 Fig. The contribution of *cis* and *trans* effects for dioxygenase gene expression variation among pairwise strain comparisons.** A heatmap (center right; scale, bottom left) shows log$_2$

fold change (log$_2$FC) values for dioxygenase (DOG) genes between parents (a combination of *cis* and *trans* effects, denoted "*cis* + *trans*") and for ASE in F1s (which correspond to *cis* effects); approximations for the magnitude of *trans* contributions to fold changes were obtained by subtracting *cis* from *cis* + *trans* log$_2$FC values (log$_2$FC values are given internal to each cell; where significant differential expression between parents was observed, or *cis* and *trans* effects were significant, asterisks are used, see Methods and S4 Table). The three groupings are as indicated at the top, and the pairs of strains for the four comparisons are indicated at bottom. Genes are ordered based on a neighbor-joining phylogenetic tree as indicated on the left. For inclusion of a DOG in the analysis, a genetic mode of control had to have been assigned in at least one comparison. Where information for a gene was not available in a genotypic comparison, cells are colored in gray.
(EPS)

**S4 Fig. The contribution of *cis* and *trans* effects for lipocalin gene expression variation among pairwise strain comparisons.** A heatmap (center right; scale, bottom left) shows log$_2$ fold change (log$_2$FC) values for lipocalin genes between parents (a combination of *cis* and *trans* effects, denoted "*cis* + *trans*") and for ASE in F1s (which correspond to *cis* effects); approximations for the magnitude of *trans* contributions to fold changes were obtained by subtracting *cis* from *cis* + *trans* log$_2$FC values (log$_2$FC values are given internal to each cell; where significant differential expression between parents was observed, or *cis* and *trans* effects were significant, asterisks are used, see Methods and S4 Table). The three groupings are as indicated at the top, and the pairs of strains for the four comparisons are indicated at bottom. Genes are ordered based on a neighbor-joining phylogenetic tree as indicated on the left. For inclusion of a lipocalin in the analysis, a genetic mode of control had to be assigned in at least one comparison. Where information for a gene was not available in a genotypic comparison, cells are colored in gray.
(EPS)

**S5 Fig. The contribution of *cis* and *trans* effects for short-chain dehydrogenase/reductase gene expression variation among pairwise strain comparisons.** A heatmap (center right; scale, bottom left) shows log$_2$ fold change (log$_2$FC) values for short-chain dehydrogenases/reductase (SDR) genes between parents (a combination of *cis* and *trans* effects, denoted "*cis* + *trans*") and for ASE in F1s (which correspond to *cis* effects); approximations for the relative magnitude of *trans* contributions to fold changes were obtained by subtracting *cis* from *cis* + *trans* log$_2$FC values (log$_2$FC values are given internal to each cell; where significant differential expression between parents was observed, or *cis* and *trans* effects were significant, asterisks are used, see Methods and S4 Table). These groupings are as indicated at the top, and the pairs of strains for the four comparisons are indicated at bottom. Genes are ordered based on a neighbor-joining phylogenetic tree as indicated on the left. For inclusion of a SDR in the analysis, a genetic mode of control had to be assigned in at least one comparison. Where information for a gene was not available in a genotypic comparison, cells are colored in gray.
(EPS)

**S1 Table. Chemical compounds used in the bioassays with concentration, corresponding IRAC group, mode of action (MoA) and time of mortality assessment.**
(XLSX)

**S2 Table. Target site mutations present by strain.**
(XLSX)

**S3 Table. Putative intact+single copy genes in pairwise comparisons.**
(XLSX)

**S4 Table. Regulatory modes for gene expression variation in pairwise comparisons.**
(XLSX)

**S5 Table. Assignments of genetic modes of control for *trans* and *cis* at two adjusted *p*-value cutoffs.**
(XLSX)

**S6 Table. Genes in the detoxification/host/peptidase (D+H+P) gene set.**
(XLSX)

**S7 Table. Enriched gene ontology (GO) terms for genes by strain comparison and genetic mode of control.**
(XLSX)

**S8 Table. Enriched gene ontology (GO) terms for genes with extremes of *cis* and *trans* genetic control.**
(XLSX)

**S9 Table. Gene expression inheritance inferred in pairwise comparisons.**
(XLSX)

## Acknowledgments

We thank Ibrahim Ismaeil for help with inbreeding and bioassays, and Robert Greenhalgh for assistance with managing read data.

## Author Contributions

**Conceptualization:** Andre H. Kurlovs, Tim De Meyer, Richard M. Clark, Thomas Van Leeuwen.

**Data curation:** Andre H. Kurlovs, Meiyuan Ji, Marilou Vandenhole.

**Formal analysis:** Andre H. Kurlovs, Berdien De Beer, Meiyuan Ji, Marilou Vandenhole.

**Funding acquisition:** Richard M. Clark, Thomas Van Leeuwen.

**Investigation:** Andre H. Kurlovs, Berdien De Beer, Meiyuan Ji, Marilou Vandenhole.

**Methodology:** Andre H. Kurlovs, Richard M. Clark, Thomas Van Leeuwen.

**Project administration:** Richard M. Clark, Thomas Van Leeuwen.

**Resources:** Tim De Meyer, Richard M. Clark, Thomas Van Leeuwen.

**Software:** Andre H. Kurlovs, Meiyuan Ji, Richard M. Clark.

**Supervision:** René Feyereisen, Richard M. Clark, Thomas Van Leeuwen.

**Validation:** Richard M. Clark, Thomas Van Leeuwen.

**Visualization:** Andre H. Kurlovs, Berdien De Beer, Meiyuan Ji.

**Writing – original draft:** Andre H. Kurlovs, Berdien De Beer, Meiyuan Ji, Marilou Vandenhole, Richard M. Clark, Thomas Van Leeuwen.

**Writing – review & editing:** Andre H. Kurlovs, Berdien De Beer, Meiyuan Ji, Marilou Vandenhole, Tim De Meyer, René Feyereisen, Richard M. Clark, Thomas Van Leeuwen.

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
