## [Decision Letter · Decision Letter 0]

8 Aug 2022

Dear Dr De Beer,

Thank you very much for submitting your Research Article entitled 'Abundant trans-driven variation in detoxification gene expression in the extreme generalist herbivore Tetranychus urticae' to PLOS Genetics.

The manuscript was fully evaluated at the editorial level and by independent peer reviewers. The reviewers agree that this is an important paper that provides significant new insights into the regulation of resistance genes. I look forwards to receiving a revised manuscript that addresses the reviewer's comments - in particular several useful comments made by Reviewer 3. Based on the reviews, we will not be able to accept this version of the manuscript, but we would be willing to review a much-revised version. We cannot, of course, promise publication at that time.

If you decide to revise the manuscript for further consideration at PLOS Genetics, please aim to resubmit within the next 60 days, unless it will take extra time to address the concerns of the reviewers, in which case we would appreciate an expected resubmission date by email to plosgenetics@plos.org.

[LINK]

Please do not hesitate to contact us if you have any concerns or questions.

Yours sincerely,

Chris Bass

Guest Editor

PLOS Genetics

Gregory P. Copenhaver

Editor-in-Chief

PLOS Genetics

Reviewer's Responses to Questions

**Comments to the Authors:**

Reviewer #1: The research paper by Kurlovs et al, entitled: “Abundant trans-driven variation in detoxification gene expression in the extreme generalist herbivore Tetranychus urticae” is a very interesting and comprehensive study on the regulatory mechanisms of gene expression in the major agricultural pest T. urticae. It provides important insights on the genetic basis of in xenobiotic resistance and plant adaptation mechanisms in T. urticae, and determines gene expression regulatory mechanisms that could be adaptable to additional agricultural pests. The methodological approach is sound, conclusions are supported by the study’s findings and the manuscript is very well written. I really have only minor comments to make that the author could consider for their revises manuscript: (1) the results of the study indeed highlight the importance of detoxification (and other) gene expression and their participation in the very high resistance levels encountered. in this respect it would be interesting to discuss the “translational” potential of these findings in operational terms; in other words, would it be possible to develop a molecular diagnostic that could be used for monitoring of these important mechanisms in field populations? (2) In line with my previous comment, how could these findings be “translated” to the development of new active compounds? (3) Since trans regulation seems to be the key mechanism for gene expression regulations it would be interesting to discuss in more details which these possible mechanisms could be (common transcription factors, epigenetic regulation- miRNAs etc).

Reviewer #2: Comments have been included as an attachment to improve readability.

Reviewer #3: The paper by Kurlovs et al. targets an important question in insect genomics, “how transcriptional changes are regulated”? The presented work looks to me quite pioneering in many aspects and therefore, I do think that the paper should be eventually published in PLOS Genetics. However, at least to my opinion, I think that more bioinformatic work should be conducted and the paper should also go through a process of condensation and focusing. Below I outline 8 major comments followed by few additional comments.

General/major comments:

I. I have read the abstract and introduction several times and still could not figure out what exactly is the scientific question. Is it general transcription regulation in Tetranychus urticae (how the genome of this spider mite works?), is it how transcription is regulated in response to acaricides selection works, to plant hosts selection? Is it only a CYP, DOG paper? What exactly are we studying so one can judge if the right experimental system was established.

II. While “cis” regulation is quite easy to understand, “trans” regulation is not. I think the authors should do much better work in the introduction to explain what “trans” regulation is all about (up to their mode of action). This cannot wait until the discussion because it significantly interferes with the understanding of the results part.

III. Methods section lines 652-667 (mite strains) raises few questions. We learn that three strains (maybe even four) have a history of selection to acaricides but we don’t know if this selection continues or not. We also do not know how long the strains were maintained on bean plants (since their collection or only before the experiment?). These two are concerns because we do not know exactly what the authors are looking for and if comparing genomes that were heavily selected versus ones that were not is the right way to go for asking questions on how genomes work?

IV. According to lines 664-665 it can be understood that each inbred strain is from one female. How many inbred strains per parent strain? How did you control for genetic drift? How many inbred strains were used per parent strain in performance and expression assays? This unclarity comment goes throughout the manuscript. Meaning, when we read that DNA was isolated from 400 adult females (line 691), are they all progeny of one female? Similarly, for the crossing experiments, were all 40 males and 40 females of each inbred strain from the same mother. If yes, how this can match the variance in the parental strain, if no, how was the work conducted?

V. Two paragraphs in the results (177-221 and 223-240) do not belong (at least to my opinion) to this paper as they do not aim to understand transcription regulation or contribute to the parents-F1 comparisons for determining “cis”, “trans” or “other” regulation. I would like to note that the first aforementioned paragraph is the best I read on genomics of acaricide resistance but it simply belongs to a different story (manuscript).

VI. At the end, parents-F1 comparisons were made only using bean as a host plant (the one on which performance is best according to Figure 2B) with no acaricide treatment. This is fine but it reflects only this specific combination. We do not know if the regulation of many genes will stay similar on more challenging hosts for example (tomato) or under acaricide treatment.

VII. There is something I do not understand in the logic or basics of the work. According to Table S4, the authors had discovered a gold mine of nearly ~7,680 genes, from which 51.9% to 66.4% fell into interesting regulation categories. Instead of conducting enrichment analyses for identifying major pathways, families etc, they focus most of their efforts on a relatively short list of about 470 genes mainly from the detox families. I don’t think that studies trying to understand how genomes function should limit themselves to such a narrow observation which covers only about 6% of the analyzed genes. In some way it feels like looking for the answer only where there is light or staying in the comfort zone. To my opinion, the authors should at least provide a list of the major enriched functions/families using all the data and their generalized mode of regulation.

VIII. What also is missing to my opinion is something beyond “cis”, “trans”, dominant, which is nice but does not really give a mechanistic explanation to regulation beyond the general feeling the genomic background matters (we kind of know that already). For example, the first time the term “transcription factor” appears is in line 564 (deep in the discussion). Why not put some efforts in running co-expression network, for the identification of modules and hubs and relate some of the discussion to the observed findings and potential discovery of regulatory elements?

Other comments

General:

The paper is a bit too long. Beside considering focusing only on transcription regulation (removing the acaricide resistance and host performance chapters), please condense by “cleaning” “methods” and “discussion” parts that appear extensively in the “results” section.

Abstract

Lines 43-44 – Most readers read the abstract before reading the whole manuscript. With this respect, it is not clear what the authors mean by “trans” instead of “cis” which is explained only in the introduction.

Lines 84-86 – Shouldn’t you be sharper here? Maybe I am wrong here but it is not just changes in promoter or enhancer regions, it is the promoter or enhancer regions physically linked to the expressed allele.

Line 93 – “versus” is confusing. Do you mean “over”?

Line 96 – Isn’t it more linked to the mode of action of the trait?

Lines 108-109, 127-130 – I would be more carful here. Adaptation to insecticides and to host plants work at very different time scales and host plant selection and utilization is far more complex than surviving a lethal dose of an insecticide. I am not sure that short-term responses to a host plant (which most studies test), actually reflect meaningful evolutionary processes of host adaptation.

Line 139 – Delete “and”.

Results:

Line 172 – Aren’t there five inbred strains it total?

Lines 292-301 and also “Methods” lines 819-823. I think I understand how “cis” and “trans” were assigned. I don’t think it is explained how expression patterns to the other categories were assigned. Also, you have six categories in the “results” and seven in the “methods”.

Line 294 – As this is quite a relaxed significance level, I would also add a filter on fold-change, especially in relation to the findings in lines 315-317. You do not want “noise” to affect too much your conclusion on the dominance of “trans” regulation (you can go to 1.5-fold, although I think that 2-fold is preferable, but I would not recommend lower than that).

Discussion:

Lines 541-543 – This example is not clear.

Methods:

Line 734 – confusing with line 729. Do you refer here to collection for RNA extraction?

**Have all data underlying the figures and results presented in the manuscript been provided?**

Reviewer #1: None

Reviewer #2: Yes

Reviewer #3: Yes

PLOS authors have the option to publish the peer review history of their article (what does this mean?). If published, this will include your full peer review and any attached files.

Reviewer #1: No

Reviewer #2: No

Reviewer #3: No

---

## [Decision Letter · Decision Letter 1]

31 Oct 2022

Dear Dr De Beer

We are pleased to inform you that your manuscript entitled "Trans-driven variation in expression is common among detoxification genes in the extreme generalist herbivore Tetranychus urticae" has been editorially accepted for publication in PLOS Genetics. Congratulations!

Please address the minor request from Reviewer 3 in the final draft you prepare for the production team (the editorial team will not need to re-evaluate).

Yours sincerely,

Chris Bass

Guest Editor

PLOS Genetics

Gregory P. Copenhaver

Editor-in-Chief

PLOS Genetics

Comments from the reviewers (if applicable):

Reviewer's Responses to Questions

**Comments to the Authors:**

Reviewer #1: The authors have adequately addressed all criticisms raised during the first round of revisions and have considerably improved their manuscript.

Reviewer #2: The authors have adequately addressed all concerns raised by the reviewer. Congratulations on your success.

Reviewer #3: Overall. as I wrote before, the paper is interesting, even pioneering and brings an important contribution to the field. Moreover, I think that the authors did a good job incorporating the reviewers’ comments after reflection. The revision was a very nice balance between change in the face of reasonable comments and standing their ground over differences of opinions.

I have only one comment on clarification that should be made.

In my previous review I commented that it is not clear “how many inbred strains per parent strain were analyzed in the performance and expression assays?” In their response, the authors explained how an inbreeding strain was produced but to my best understanding did not answer my question.

I think it is important to clarify if each parent strain was eventually represented by one inbred strain that started from one female, or multiple inbred lines. If only one, it might be hard to argue that it represents the parental strain genetic background.

**Have all data underlying the figures and results presented in the manuscript been provided?**

Reviewer #1: Yes

Reviewer #2: Yes

Reviewer #3: Yes

PLOS authors have the option to publish the peer review history of their article (what does this mean?). If published, this will include your full peer review and any attached files.

Reviewer #1: No

Reviewer #2: No

Reviewer #3: No

**Data Deposition**

http://datadryad.org/submit?journalID=pgenetics&manu=PGENETICS-D-22-00819R1

**Press Queries**

---

## [Editor Report · Acceptance letter]

8 Nov 2022

PGENETICS-D-22-00819R1 

Trans-driven variation in expression is common among detoxification genes in the extreme generalist herbivore Tetranychus urticae 

Dear Dr De Beer, 

We are pleased to inform you that your manuscript entitled "Trans-driven variation in expression is common among detoxification genes in the extreme generalist herbivore Tetranychus urticae" has been formally accepted for publication in PLOS Genetics! Your manuscript is now with our production department and you will be notified of the publication date in due course.

With kind regards,

Anita Estes

PLOS Genetics

On behalf of:
